# Age-related islet inflammation marks the proliferative decline of pancreatic beta-cells in zebrafish

Sharan Janjuha[1,2,3†], Sumeet Pal Singh[1†], Anastasia Tsakmaki[4],
S Neda Mousavy Gharavy[5,6], Priyanka Murawala[1], Judith Konantz[1], Sarah Birke[1],
David J Hodson[7,8], Guy A Rutter[5,6], Gavin A Bewick[4], Nikolay Ninov[1,2,3]*

[1]DFG-Center for Regenerative Therapies Dresden, Technische Universität Dresden, Dresden, Germany; [2]Paul Langerhans Institute Dresden, Helmholtz Zentrum München at the University Hospital, German Center for Diabetes Research (DZD e. V.), Dresden, Germany; [3]Faculty of Medicine Carl Gustav Carus, Technische Universität Dresden, German Center for Diabetes Reseach (DZD e.V.), Dresden, Germany; [4]Diabetes Research Group, School of Life Course Sciences, Faculty of Life Sciences & Medicine, King's College London, London, United Kingdom; [5]Section of Cell Biology and Functional Genomics, Division of Diabetes, Endocrinology, and Metabolism, Imperial College London, London, United Kingdom; [6]Consortium for Islet Cell Biology and Diabetes, Department of Medicine, Imperial College London, London, United Kingdom; [7]Centre for Endocrinology, Diabetes, and Metabolism, University of Birmingham, Edgbaston, United Kingdom; [8]Institute of Metabolism and Systems Research, University of Birmingham, Edgbaston, United Kingdom

*For correspondence:
nikolay.ninov@tu-dresden.de

[†]These authors contributed equally to this work

**Competing interests:** The authors declare that no competing interests exist.

**Abstract** The pancreatic islet, a cellular community harboring the insulin-producing beta-cells, is known to undergo age-related alterations. However, only a handful of signals associated with aging have been identified. By comparing beta-cells from younger and older zebrafish, here we show that the aging islets exhibit signs of chronic inflammation. These include recruitment of *tnfα*-expressing macrophages and the activation of NF-kB signaling in beta-cells. Using a transgenic reporter, we show that NF-kB activity is undetectable in juvenile beta-cells, whereas cells from older fish exhibit heterogeneous NF-kB activity. We link this heterogeneity to differences in gene expression and proliferation. Beta-cells with high NF-kB signaling proliferate significantly less compared to their neighbors with low activity. The NF-kB signaling[hi] cells also exhibit premature upregulation of *socs2*, an age-related gene that inhibits beta-cell proliferation. Together, our results show that NF-kB activity marks the asynchronous decline in beta-cell proliferation with advancing age.
DOI: https://doi.org/10.7554/eLife.32965.001

## Introduction

Aging is a universal process that detrimentally changes the characteristics of cells in multicellular organisms. A hallmark of aging is the reduction in cellular renewal and proliferation across different tissues and organs (*Yun, 2015*). The insulin producing beta-cells, which reside in the islets of Langerhans, provide a good model to study regulators of cellular aging. Whereas young beta-cell are highly proliferative and increase rapidly in number from the prenatal phase until early stages of development in mammals, beta-cell proliferation becomes dramatically reduced in adults (*Perl et al., 2010*). Nevertheless, adult beta-cell proliferation can increase under specific conditions such as obesity and pregnancy (*Parsons et al., 1992*; *Weir et al., 2001*). It remains unclear whether this proliferation is

restricted to a privileged population of beta-cells that retain replicative potential even in adult life, or whether it represents stochastic cell cycle re-entry.

Previous studies have indicated that both extrinsic factors, such as the vasculature, and intrinsic factors, such as chromatin modifications, may influence the age-related changes in beta-cells. For example, rejuvenating the beta-cell environment by implanting old islets in younger animals is sufficient to restore the proliferative potential of the aged beta-cells (*Almaça et al., 2014*; *Salpeter et al., 2013*). In addition, transcriptome and methylome studies revealed age-dependent DNA methylation changes at cell-cycle regulators, which may contribute to the quiescence of aging beta-cells (*Avrahami et al., 2015*; *Arda et al., 2016*). Furthermore, analysis of gene expression in islets from aging mice showed an age-dependent decline of transcripts encoding the platelet-derived growth factor-receptors Pdgfra and Pdgfrb as well as its ligand Pdgf. This decline in expression was shown to underlie a decline in beta-cell proliferation with aging (*Chen et al., 2011*). Likewise, the expression of the transcription factor FoxM1 declines with aging and the forced expression of its activated form in aged beta-cells is sufficient to re-ignite replication (*Golson et al., 2015*). In addition, the prostaglandin receptors (E-Prostanoid Receptor 3 and 4) might also regulate beta-cell proliferation in an age-dependent manner (*Carboneau et al., 2017*).

An important aspect of beta-cell biology is the presence of significant heterogeneity within a seemingly homogenous collection of cells. In particular, beta-cells within the islet and between islets may belong to subpopulations with different 'ages' (*van der Meulen et al., 2017*; *Singh et al., 2017*), with the proportion of young-to-old beta-cells changing with the age of the animal (*Aguayo-Mazzucato et al., 2017*). In addition, recent studies have identified various markers of beta-cell heterogeneity such as *Fltp*, ST8SIA1 and CD9 (*Bader et al., 2016*; *Dorrell et al., 2016*). Specifically, *Fltp* was shown to distinguish the proliferative beta-cells from the more functional ones. However, the markers of beta-cell heterogeneity have not yet been shown to play a direct role in establishing phenotypic differences among the beta-cell subpopulations. In addition, it remains unclear how aging shapes the proliferative heterogeneity of the beta-cells.

To identify signals that change in beta-cells during organismal aging, we used the zebrafish as a model. We first characterized the rate of beta-cell proliferation in juvenile, younger and older adults, and found that proliferation declines with advancing age. We performed transcriptomics of beta-cells from younger and older animals, which identified an upregulation of genes involved in inflammation, including NF-kB signaling. The analysis of inflammatory signaling with single-cell resolution using a transgenic GFP reporter line confirmed that NF-kB signaling was activated in a heterogeneous manner at the level of individual beta-cells. Notably, beta-cells with higher levels of NF-kB signaling exhibit a more pronounced proliferative decline compared to their neighbors with lower activity. These cells also express higher levels of *socs2*, which can inhibit beta-cell proliferation in a cell-autonomous manner. Our work identifies NF-kB signaling as a marker of beta-cell aging and their proliferative decline.

## Results

### Beta-cell proliferation declines with advancing age in zebrafish

To monitor the endogenous rate of proliferation of zebrafish beta-cells, we used the beta-cell-specific fluorescence ubiquitination cell cycle indicator (FUCCI) lines, *Tg(ins:Fucci-G1)* and *Tg(ins:Fucci-S/G2/M)* (*Ninov et al., 2013*). The FUCCI system uses fluorescent proteins fused with cdt1 to label cells in the G0/G1 phases of cell cycle with red fluorescence and geminin to label cells in S/G2/M with green fluorescence (*Figure 1a*). We imaged whole primary islets from normally-fed fish at 35 days-post-fertilization (dpf), 3 months-post-fertilization (mpf) and 1 year-post-fertilization (ypf) (*Figure 1c–e*). We calculated the percentage of *Tg(ins:Fucci-G1)*-negative and *Tg(ins:Fucci-S/G2/M)*-positive cells among the total number of beta-cells per islet. We found that the percentage of proliferating beta-cells declined with advancing age. Whereas in islets from 35 dpf animals, on average 1.53% ± 0.72 (n = 5) of the beta-cells were proliferating, this number was reduced to 0.15% ± 0.07 (n = 9) and 0.06% ± 0.02 in islets from 3 mpf and 1 ypf animals, respectively (n = 10) (*Figure 1b*). A similar decline in beta-cell proliferation was observed also in the secondary islets, which arise from the differentiation of *sox9b*-expressing progenitors lining the pancreatic ducts (*Figure 1—figure supplement 1a–d*).

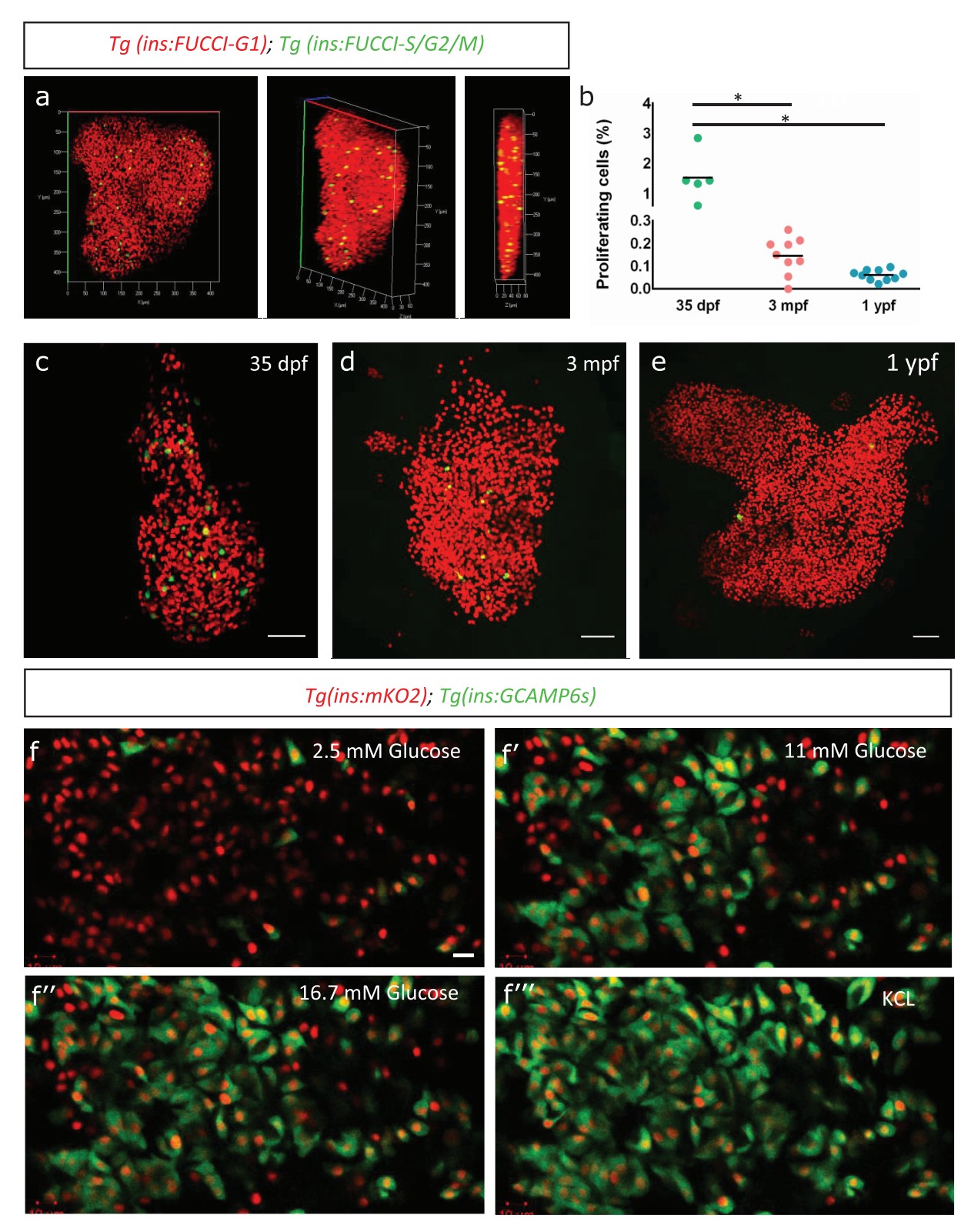

**Figure 1.** Beta-cell proliferation declines with age. (**a**) 3D-rendering of a primary islet from *Tg(ins:Fucci-G1);Tg(ins:Fucci-S/G2/M)* animals at 3 mpf showing nuclear *Tg(ins:Fucci-G1)* (red) and *Tg(ins:Fucci-S/G2/M)* (green) expression. (**b**) Quantification of percentage of *Tg(ins:Fucci-S/G2/M)*-positive and *Tg(ins:Fucci-G1)*-negative (green-only) beta-cells at 35 dpf (n = 5), 3 mpf (n = 9) and 1 ypf (n = 10) animals. Each dot represents one animal. Horizontal bars represent mean values (one-way ANOVA, *p<0.05). (**c, d, e**) Confocal projection of whole-mount islets from *Tg(ins:Fucci-G1);Tg(ins:*

*Figure 1 continued*

*Fucci-S/G2/M)* animals at 35 dpf, 3 mpf and 1 ypf. Anterior to the top. Scale bar 50 μm. (**f**) Ex vivo live-imaging of beta-cells from *Tg(ins:nlsRenilla-mKO2)*;Tg(ins:GCaMP6s) animals at 3 mpf. Beta-cells (red) were stimulated with 2.5 (basal) mM D-Glucose, (**f'**) 11 mM D-glucose, (**f''**) 16.7 mM D-glucose and (**f'''**) depolarized using 30 mM KCl while monitoring GCAMP6s-fluorescence (green). Scale bar 10 μm.

DOI: https://doi.org/10.7554/eLife.32965.002

The following figure supplement is available for figure 1:

**Figure supplement 1.** Beta-cell proliferation declines with age in secondary islets.

DOI: https://doi.org/10.7554/eLife.32965.003

To confirm that adult beta-cells within the zebrafish primary islets are functional, we analyzed glucose-stimulated calcium influx using *Tg(ins:GCaMP6s)* transgenic line, a genetically encoded calcium indicator that binds to increasing intracellular $Ca^{2+}$ and emits green fluorescence (*Singh et al., 2017*). We crossed this line to *Tg(ins:nlsRenilla-mKO2)*, which marks the beta-cells with red fluorescence. This double transgenic system allowed us to visualize the response of beta-cells to increasing concentrations of glucose over time ex vivo (n = 10) (*Figure 1f–f'''*). We found that adult beta-cells were sensitive to glucose, as beta-cells exhibited calcium influx upon stimulation with increasing glucose concentrations.

## Aging is associated with transcriptional changes in zebrafish beta-cells

To determine changes in gene expression in beta-cells with increasing age, we used fluorescence-activated cell sorting (FACS) coupled with next-generation RNA-Sequencing to profile fluorescently labeled beta-cells from 3 mpf and 1 ypf animals (*Figure 2a–a'*, *Figure 2—figure supplement 1a*). We selected these two stages in order to avoid confounding changes in gene expression associated with the morphogenesis and the remodeling of the islets occurring during the juvenile stages (*Singh et al., 2017*). Thus, we compared the transcriptomes of beta-cells at 3 mpf and 1 ypf to identify genes that increase in expression with increasing age. In order to avoid introducing sequencing noise or bias, RNA-sequencing of sorted beta-cells was carried out without PCR amplification of the staring m RNA. A comparison between beta-cells from 3 mpf and 1 ypf animals revealed 74 genes that showed 1.5-$\log_2$fold difference (p<0.05) in expression (*Figure 2b*), of which 61 genes were upregulated and 13 genes were downregulated in older beta-cells (*Supplementary file 1*). Literature survey and unbiased gene ontology analysis using DAVID (*Huang et al., 2009a*; *Huang et al., 2009b*) revealed that the upregulated genes were involved in the negative regulation of growth-factor signaling including *socs2*, *cish*, *spry4* and *fstl1* (*Figure 2c–c'*). We also found upregulation of genes involved in ER stress including *trib3* and *cebpd*, as well as genes associated with increased risk of developing Type two diabetes and glucose intolerance (*prtfa*, *lpp* and *socs2*) (*Fang et al., 2014*; *Szabat et al., 2016*; *Kato et al., 2006*; *Lebrun et al., 2010*; *Nair et al., 2014*; *Liu et al., 2008*).

## NF-kB-signaling is activated heterogeneously in the beta-cells with advancing age

In addition to the genes involved in regulating proliferation and ER stress, cytokine-mediated signaling was over-represented in the gene ontology analysis performed using DAVID (*Figure 2c'*). We found that transcripts associated with an inflammatory signature, such as interleukins, complement factors and members of the NF-kB pathway, including *il15*, *c9*, *tnfrsf1b*, *cd74a*, *cd74b* (*Starlets et al., 2006*), also increased in expression in islets from older animals (*Supplementary file 1*). Specifically, *tnfrsf1b* belongs to a superfamily of cytokine receptors, which respond to Tumor Necrosis Factor (TNF) and activate NF-kB, an inducible and ubiquitous transcription factor that senses inflammation (*Espı́nEspín-Palazó́nPalazón et al., 2014*). In order to validate the changes in gene-expression of *tnfrsf1b* at the level of individual cells, we performed single-cell RT-qPCR of sorted beta-cells (*Supplementary file 2*). Notably, the single-cell RT-qPCR revealed that there was an increase in the proportion of beta-cells expressing *tnfrsf1b* in islets of older animals (*Figure 2d*). This was also true for additional components of the NF-kB pathway, including *ikbaa* and *tnfα.* In contrast, the proportion of sorted cells expressing known beta-cell markers such as *insulin*, *islet1* and *neurod1*, remained similar (*Figure 2d*).

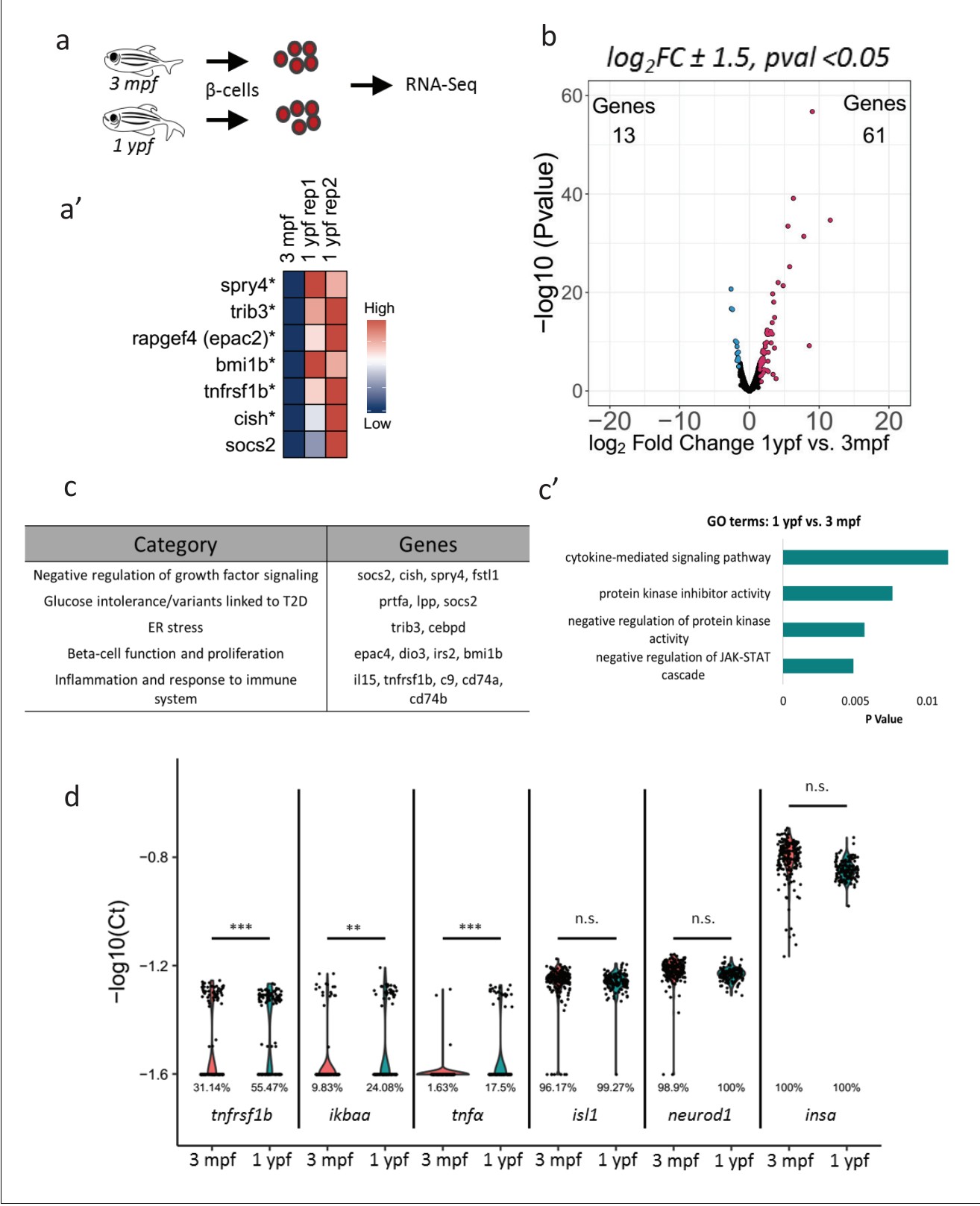

**Figure 2.** Transcriptome profiling of younger and older beta-cells. (a) Schematic showing isolation and FAC-sorting of beta-cells from *Tg(ins:nlsRenilla-mKO2)* animals at 3 mpf and 1 ypf followed by high-throughput mRNA-Sequencing. (a') Heatmap depicting differentially regulated genes among the beta-cells at 1 ypf and 3 mpf involved in beta-cell proliferation, function and inflammation (asterisks denote genes vaildated by single-cell RT-qPCR). (b) Volcano plot representing the distribution of genes that were differentially regulated in beta-cells from 1 ypf and 3 mpf (1.5-log2fold change, p<0.05). *Figure 2 continued on next page*

*Figure 2 continued*

(c) The biological categories of enriched genes in beta-cells at one ypf (1.5-log$_2$fold change, p<0.05) based on literature survey. (c') Unbiased gene-ontology analysis using DAVID of genes enriched in beta-cells at 1 ypf (p<0.05). (d) Gene expression analysis was carried out using single-cell RT-qPCR. Violin plots denote expression distribution of the candidate genes. The Y-axis shows -log$_{10}$(Ct) values of transcript levels in single beta-cells. The X-axis shows gene names and the respective developmental stages. The percentage values under each violin plot denote the proportion of beta-cells with detectable transcript levels. The cycle threshold for detectable gene expression was set as Ct = 40. The value $-1.6$ (-log$_{10}$(40)) on the Y-axis represents undetectable expression as measured by single-cell RT-qPCR (see Materials and methods). Each dot represents one beta-cell. Significance testing for differences in proportion of cells with detectable gene expression at each stage was performed using Pearson's Chi-Square test (**p<0.01, ***p<0.001).
DOI: https://doi.org/10.7554/eLife.32965.004

The following figure supplement is available for figure 2:

**Figure supplement 1.** Fluorescent activated cell sorting of beta-cells.
DOI: https://doi.org/10.7554/eLife.32965.005

We then wanted to test if overexpressing *tnfrsf1b* in beta-cells can induce NF-kB signaling. To do so, we cloned *tnfrsf1b* in a plasmid containing an upstream insulin promoter and injected it into one-cell-stage embryos. The ensuing stochastic genomic integration and expression from the insulin promoter leads to mosaic overexpression of *tnfrsf1b* specifically in beta-cells. We analyzed the activity of NF-kB using an NF-kB signaling reporter line, *Tg(NF-kB:GFP)* (*Kanther et al., 2011*). This reporter expresses GFP under the control of six tandem NF-kB-binding sites, such that GFP is expressed upon the nuclear translocation and binding of NF-kB dimer to the NF-kB binding sites. We saw that a higher proportion of beta-cells from animals injected with *ins:tnfrsf1b* expressed GFP at 5 dpf compared to controls. A total of 32.2% ± 32.07 beta-cells (n = 6) in the *tnfrsf1b*-injected animals expressed GFP as compared to 2.4% ± 1.98 beta-cells (n = 5) in the controls (*Figure 3a–b*).

The ability of *tnfrsf1b* overexpression to activate NF-kB signaling and the increase in the proportion of beta-cells that upregulate *tnfrsf1b* with age (*Figure 2d*) prompted us to follow-up on the endogenous levels of NF-kB signaling in the beta-cells. We performed a temporal analysis of NF-kB activity in beta-cells by imaging the islets from *Tg(NF-kB:GFP)* animals (*Figure 3c–e*, *Figure 3—figure supplement 1a–d*). We found that GFP intensity was too low to be detected in beta-cells from the primary or secondary islets of juveniles (1 mpf) (*Figure 3c*, *Figure 3—figure supplement 1a*). In contrast, beta-cells from 3 mpf animals exhibited a detectable, salt-and-pepper pattern of GFP expression (*Figure 3d*, *Figure 3—figure supplement 1b*), suggesting heterogeneous NF-kB activation, which is consistent with the heterogeneous expression of *tnfrsf1b* (*Figure 2d*). Notably, nearly all beta-cells in both the primary and secondary islets from 1 ypf animals express GFP (*Figure 3e*, *Figure 3—figure supplement 1c*).

To better quantify the proportions of GFP-positive cells in younger and older islets, we labeled beta-cells from 3 mpf and 1 ypf *Tg(NF-kB:GFP)* animals using the Zn$^{2+}$ chelator TSQ (*Kim et al., 2000*), which preferentially labels beta-cells due to their high-zinc content (*Figure 3—figure supplement 2a–b*). TSQ-labeled beta-cells were then passed through FACS and were analyzed for the levels of GFP expression in each cell. Flow cytometry analysis of cells from 3 mpf and 1 ypf animals confirmed the presence of two populations at each stage based on GFP-fluorescence intensity (*Figure 3f–g*) (n = 10). Quantifying the proportion of cells within the low- and high-GFP expressing regions indicated that a higher proportion of cells express GFP in older animals (*Figure 3f–g*). Thus, we found that not only does the overall GFP expression increases in individual cells with increasing age, but a higher proportion of cells with GFP expression were present in the islets of the older compared to younger animals (*Figure 3f–g*).

In order to verify that the increase in GFP levels in older fish is not simply due to the accumulation of GFP protein, we quantified using RT-qPCR the differences in GFP mRNA in beta-cells sorted from 3 mpf and 1 ypf animals of the genotype *Tg(ins:mCherry);Tg(NF-kB:GFP)*. We saw approximately 50% increase in the GFP transcript levels in beta-cells from 1 ypf animals as compared to 3 mpf animals (*Figure 3—figure supplement 3a*). This result corroborates the increase in NF-kB reporter activity in beta-cells between the two time points. Furthermore, we used index sorting of single-cells, which allows to correlate transcript levels with GFP fluorescence intensity in individual beta-cells. Overall, there was a positive correlation between GFP mRNA and GFP fluorescence intensity across cells (R$^2$ = 0.28) (*Figure 3—figure supplement 3b*).

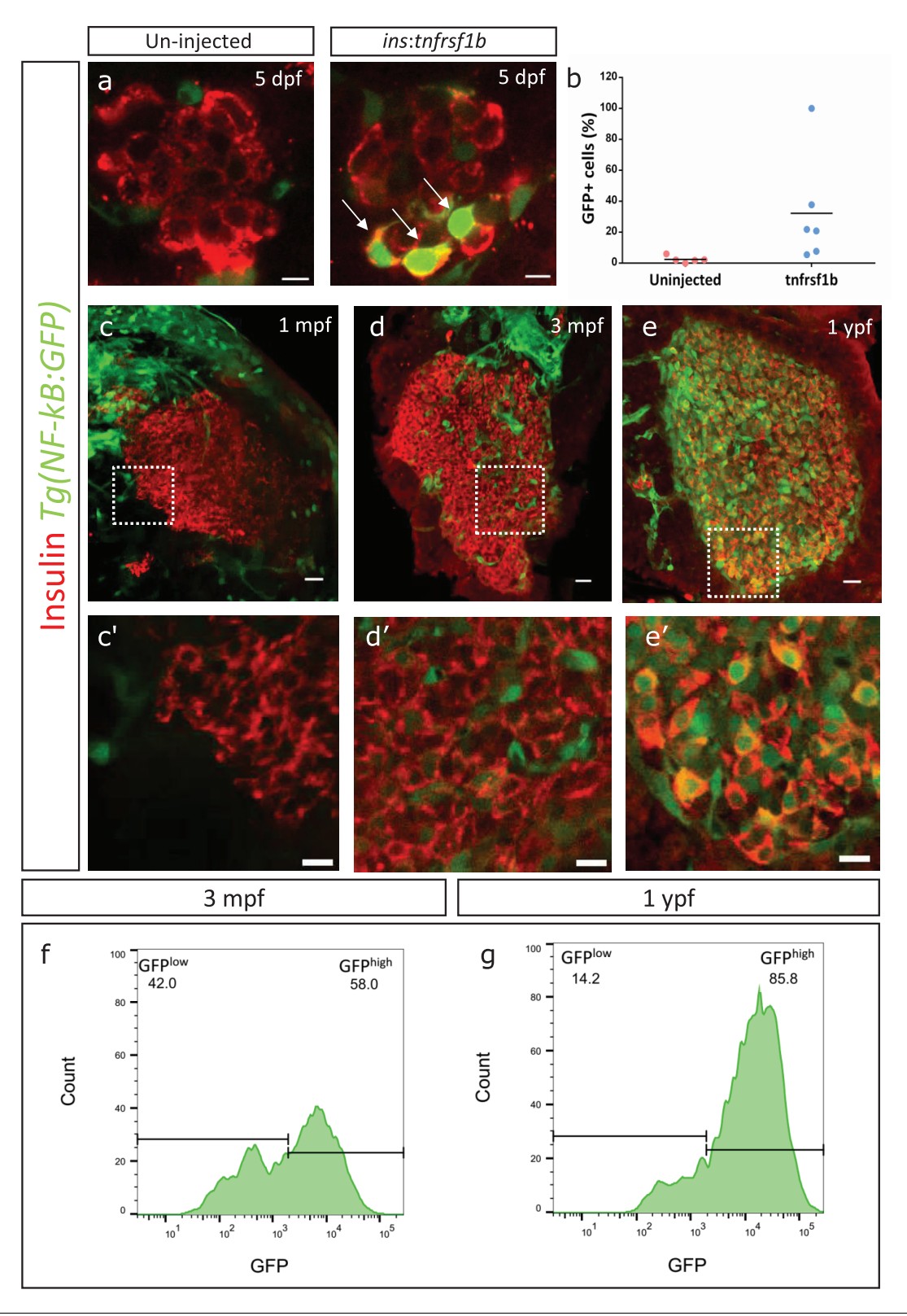

**Figure 3.** An inflammation reporter reveals heterogeneous activation of NF-kB signaling in beta-cells with age. (**a**) The images show single confocal planes from islets of 5 dpf larvae. The *tnfrsf1b* coding sequence was expressed under the control of the insulin promoter. The plasmid was injected in *Tg(NF-kB:GFP)* embryos at the one-cell-stage, leading to mosaic and stochastic expression of the construct in beta-cells. The *Tg(NF-kB:GFP)* reporter expresses GFP (green) under the control of six tandem repeats of NF-kB DNA-binding sites. Beta-cells were labelled using an insulin antibody (red). *Figure 3 continued on next page*

*Figure 3 continued*

Arrows indicate GFP-positive beta-cells. Scale bar 5 µm. (**b**) The graph shows the percentage of GFP-positive and insulin-positive cells in uninjected controls (n = 5) and *tnfrsf1b* injected animals (n = 6) at 5 dpf. Horizontal bars represent mean values. (**c–e**) Confocal stack of islets from *Tg(NF-kB:GFP)* animals at 1 mpf, 3 mpf and 1 ypf. Beta-cells were labeled using an insulin antibody (red). *NF-kB*:GFP reporter expression is shown in green. Scale bars 20 µm. (**c'–e'**) Insets show high magnification single planes of the confocal stacks (corresponding to the regions shown using white dotted-lines in the top panels). Scale bar 10 µm. (**f–g**) Beta-cells from 3 mpf *Tg(NF-kB:GFP)* animals were labeled with TSQ (Zn2+ labeling dye) and analyzed using FACS. The graph shows GFP intensity (along the X-axis) and the distribution of beta-cells at 3 mpf and 1 ypf. Horizontal lines indicate the division point between GFP$^{low}$ and GFP$^{high}$ levels. Percentage values represent proportion of cells with GFP$^{low}$ or GFP$^{high}$ expression.

DOI: https://doi.org/10.7554/eLife.32965.006

The following figure supplements are available for figure 3:

**Figure supplement 1.** Activation of NF-kB signaling in beta-cells of the secondary islets with age.

DOI: https://doi.org/10.7554/eLife.32965.007

**Figure supplement 2.** Fluorescent activated cell sorting of *NF-kB*:GFP$^{high}$ and *NF-kB*:GFP$^{low}$ beta-cells.

DOI: https://doi.org/10.7554/eLife.32965.008

**Figure supplement 3.** *NF-kB:EGFP* mRNA levels in beta-cells increase with age.

DOI: https://doi.org/10.7554/eLife.32965.009

## Immune cells infiltrate the islet during development and persist throughout adult life

An enrichment of genes associated with an inflammatory signature in beta-cells from older fish together with the heterogeneous activation of the NF-kB pathway prompted us to look for additional signs of islet inflammation. One cell type important for the response and resolution of inflammation is the tissue-resident macrophage. To study this cell type in the developing islet, we labeled immune cells using a pan-leukocyte marker, L-plastin, which marks the monocyte/macrophage lineage in zebrafish (*Mathias et al., 2009*). We found that whereas innate immune cells were not present in the islets during the larval stages (15–21 dpf), they had infiltrated them during the late juvenile stages (45 dpf) (*Figure 4a*). Analysis of the macrophage reporter line, *Tg(mpeg1:mCherry)*, revealed that the innate immune cells were macrophages, whereas neutrophils could not be detected, as assessed by the neutrophil-specific line *Tg(lyz2:GFP)* (*Figure 4—figure supplement 1a*, data not shown).

To test whether these infiltrating immune cells express inflammatory cytokines, such as TNFα, we made use of a *TgBAC(tnfα:GFP)* transgenic line and examined the presence of *tnfα*-expressing leukocytes within the L-plastin-positive population (*Marjoram et al., 2015*). On average 25% ± 10.9 (n = 5) and 17% ± 11.1 (n = 5) of the L-plastin-positive cells inside the islet expressed *tnfα*:GFP in 3 mpf and one ypf animals, respectively (p>0.05) (*Figure 4b*, *Figure 4—figure supplement 1b*). However, the number of *tnfα*:GFP-positive cells, as well as the total number of L-plastin-positive cells showed increasing trends in the islets from older animals (*Figure 4—figure supplement 1c–d*). Analysis of *TgBAC(tnfα:GFP)* together with specific labeling of macrophages using *Tg(mpeg1:mCherry)* confirmed that the *tnfα*:GFP-expressing leukocytes were macrophages (*Figure 4c*).

We next wanted to test whether *tnfα*-expression is capable of inducing inflammatory activity in the beta-cells. To this end, we placed *tnfα* under the insulin promoter in order to drive beta-cell-specific expression. We injected the construct in one-cell-stage *Tg(NF-kB:GFP)* embryos and analyzed GFP-expression in beta-cells at 5 dpf. Indeed, we found that *tnfα*-expression alone could induce *NF-kB*:GFP reporter activity (*Figure 4d–e*).

## *NF-kB*:GFP$^{high}$ beta-cells proliferate less compared to their neighbors

Based on the earlier observation that beta-cell proliferation declines in older fish, and the finding that *NF-kB*:GFP expression increases, we asked if high NF-kB activity and beta-cell proliferation were inversely correlated. We performed 5-ethynyl-2´-deoxyuridine (EdU) incorporation assay to mark the proliferating beta-cells in 3 mpf *Tg(NF-kB:GFP)* animals and examined the levels of *NF-kB*:GFP in the EdU-positive and negative beta-cells. We measured the normalized GFP intensity in all beta-cells in the islets of 3 mpf animals (n = 9). The total normalized mean GFP intensity of all the sections belonging to one islet, designated GFP$^{total}$, was set as a threshold for each respective islet. The beta-cells with normalized mean GFP intensity higher than GFP$^{total}$ were categorized as *NF-kB*:GFP$^{high}$ while cells with normalized mean GFP intensity lower than GFP$^{total}$ were categorized as

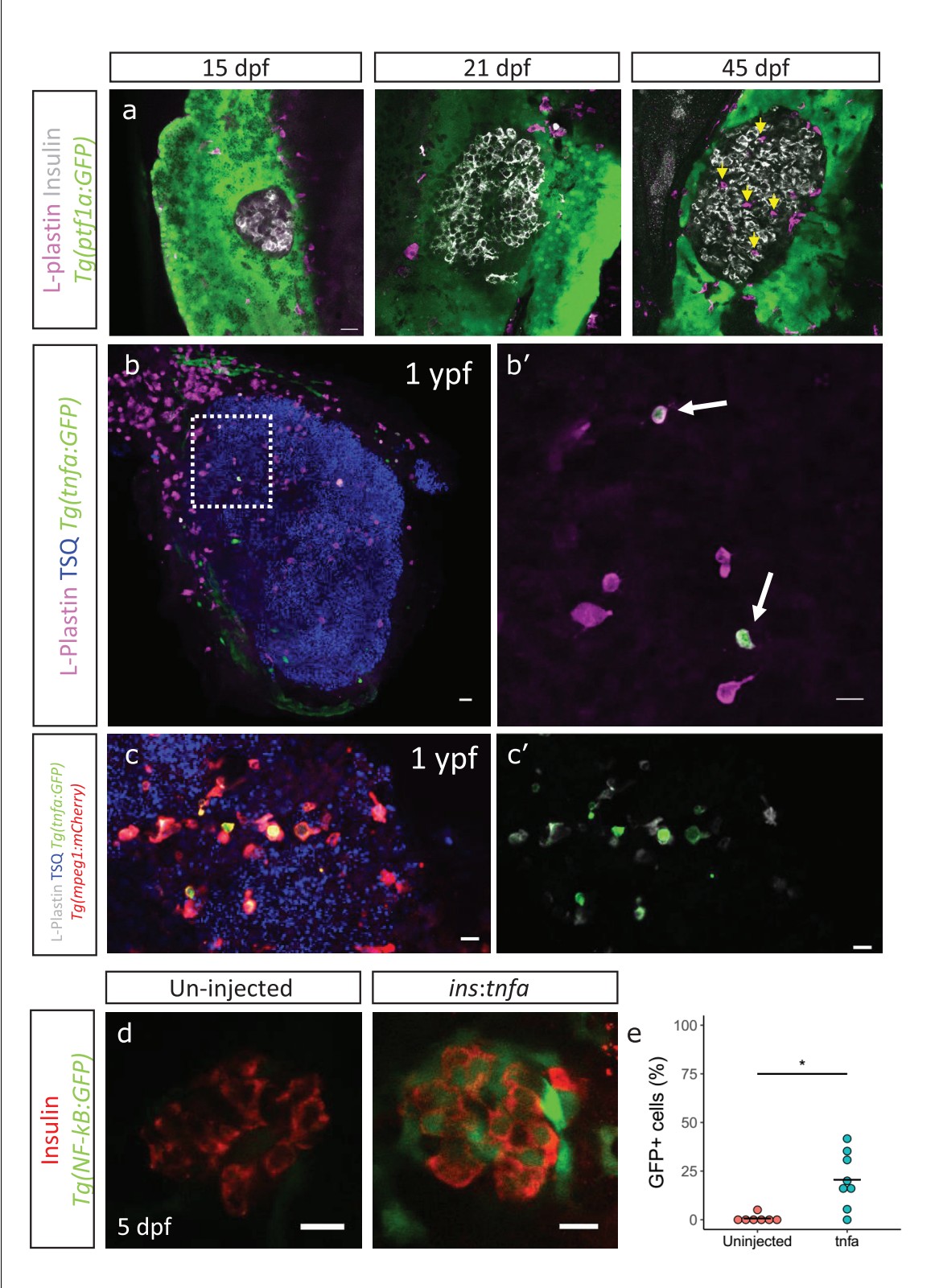

**Figure 4.** Immune cells infiltrate the islet during development and persist throughout adult life. (**a**) Confocal images of pancreata from 15, 21 and 45 dpf animals. Beta-cells were labeled using an insulin antibody (grey), leukocytes were labeled using an L-plastin antibody (magenta) and *Tg(ptf1a*:GFP) marks the acinar cells (green). Immune cells are present within the islet at 45 dpf (arrows). (**b**) Confocal images of whole islets from *Tg(tnfα:GFP)* animals at 1 ypf. Islets were labeled using TSQ (Zn$^{2+}$ labeling dye) (blue), leukocytes were labelled with an L-plastin antibody (magenta) and *Tg(tnfα*:GFP) marks

*Figure 4 continued on next page*

*Figure 4 continued*

cells expressing *tnfα* (green). Scale bars 20 μm. (**b'**) Insets show high-magnification single planes from the confocal stacks (corresponding to the area marked using a white dotted-line in b). Scale bar 10 μm. (**c–c'**) Confocal image of a one ypf islet showing a single plane. The *TgBAC(tnfα:GFP)* line marks the *tnfα*-positive cells (green), whereas *Tg(mpeg1:mCherry)* marks the macrophages (red). The L-plastin antibody marks all leukocytes (grey) and TSQ (Zn2 +labeling dye) was used to mark the islet (n = 5). Scale bar, 10 μm. (**d**) Confocal images showing islets at five dpf. The *tnfα* cDNA was expressed under the insulin promoter. The plasmid was injected in *Tg(NF-kB:GFP)* embryos at the one-cell-stage and the islets were analyzed at 5 dpf. Beta-cells were labeled with an insulin antibody (red). *Tg(NF-kB*:GFP) reporter expression is shown in green. (**e**) The graph shows the percentage of GFP and insulin double-positive cells in un-injected controls (n = 7) and *ins:tnfα* injected animals (n = 8) at five dpf. Horizontal bars represent mean values (two-tailed t-test, *p<0.05).

DOI: https://doi.org/10.7554/eLife.32965.010

The following figure supplement is available for figure 4:

**Figure supplement 1.** Immune cells infiltrate the islet during development.

DOI: https://doi.org/10.7554/eLife.32965.011

GFP$^{low}$. We observed that a lower proportion of the *NF-kB*:GFP$^{high}$ cells had incorporated EdU over a 2-day period as compared to GFP$^{low}$ cells (**Figure 5a–c**). In order to confirm that the GFP fluorescence of beta-cells remains stable over the 2-day period of EdU incorporation, we followed individual FAC-sorted *NF-kB*:GFP$^{high}$ and $^{low}$ beta-cells over 72 hr ex vivo. Indeed, the GFP fluorescence remained stable over the time-period of the experiment (**Figure 5—figure supplement 1**). In addition, to obtain a snapshot of the proliferative status of the cells, we performed immunohistochemistry for the proliferating cell nuclear antigen (PCNA), which marks proliferating cells. A higher proportion of *NF-kB*:GFP$^{low}$ cells were positive for PCNA, as compared to *NF-kB*:GFP$^{high}$ cells (**Figure 5d**, **Figure 5—figure supplement 2**). We conclude that beta-cells with high NF-kB signaling proliferate significantly less compared to their neighbors with lower activity.

## *Socs2* is enriched in *NF-kB*:GFP$^{high}$ beta-cells and inhibits proliferation

To investigate molecular factors to explain the lower proliferation of *NF-kB*:GFP$^{high}$ beta-cells, we separated the beta-cells from 3 mpf animals into GFP$^{high}$ and GFP$^{low}$ populations using a double transgenic line *Tg(ins:mCherry);Tg(NF-kB:GFP)* by FACS (**Figure 6a**, **Figure 6—figure supplement 1**). Using RT-qPCR analysis of the GFP$^{high}$ and GFP$^{low}$ populations, we then quantified the expression levels of selected candidate genes that we previously found to be significantly enriched in beta-cells from older animals (1 ypf) in the transcriptomic analysis. We found that *socs2* showed more than 2.5-fold higher expression (n = 4 biological replicates, n = 3 animals per replicate, 1000 cells per condition) in the GFP$^{high}$ cells compared to GFP$^{low}$, whereas other genes did not exhibit significantly higher expression (**Figure 6b**, **Figure 6—source data 1**).

To test if higher levels of *socs2* expression can inhibit beta-cell proliferation, we generated a bicistronic construct containing CFP linked to *socs2* via a viral T2A sequence, and placed it under the control of the insulin promoter. Injecting the plasmid in one-cell-stage zebrafish embryos leads to mosaic and stochastic expression of *socs2* in beta-cells at later stages (**Figure 6c**). To quantify the effect of *socs2* expression on proliferation, we injected the plasmid in *Tg(ins:Fucci-G1);Tg(ins:Fucci-S/G2/M)* embryos, such that beta-cells in the G0/G1 phases of cell-cycle were labeled in red, whereas cells in the S/G2/M phases of cell cycle were labeled in green. The cells expressing *socs2* were also CFP-positive, allowing us to distinguish them from wild-type beta-cells in the same islet (**Figure 6d**). We then quantified the proportion of proliferating CFP-positive and CFP-negative beta-cells at 23–25 dpf, a stage characterized by higher rates of beta-cell proliferation. We found that whereas 8.44% ± 3.37 of the CFP-negative beta-cells were proliferating, only 1.08% ± 1.65 CFP-positive beta-cells exhibited cell-cycle progression (n = 9) (**Figure 6e**). Overexpression of CFP alone, or CFP-T2A-*rapgef4* and CFP-T2A-*spry4* in this mosaic manner did not affect proliferation (**Figure 6—figure supplement 2**). Thus, *socs2* can cell-autonomously inhibit beta-cell proliferation. Altogether, these results suggest that the higher endogenous expression of *socs2* in *NF-kB*:GFP$^{high}$ compared to *NF-kB*:GFP$^{low}$ beta-cells could contribute to the proliferative heterogeneity among beta-cells based on the differences in NF-kB signaling strength.

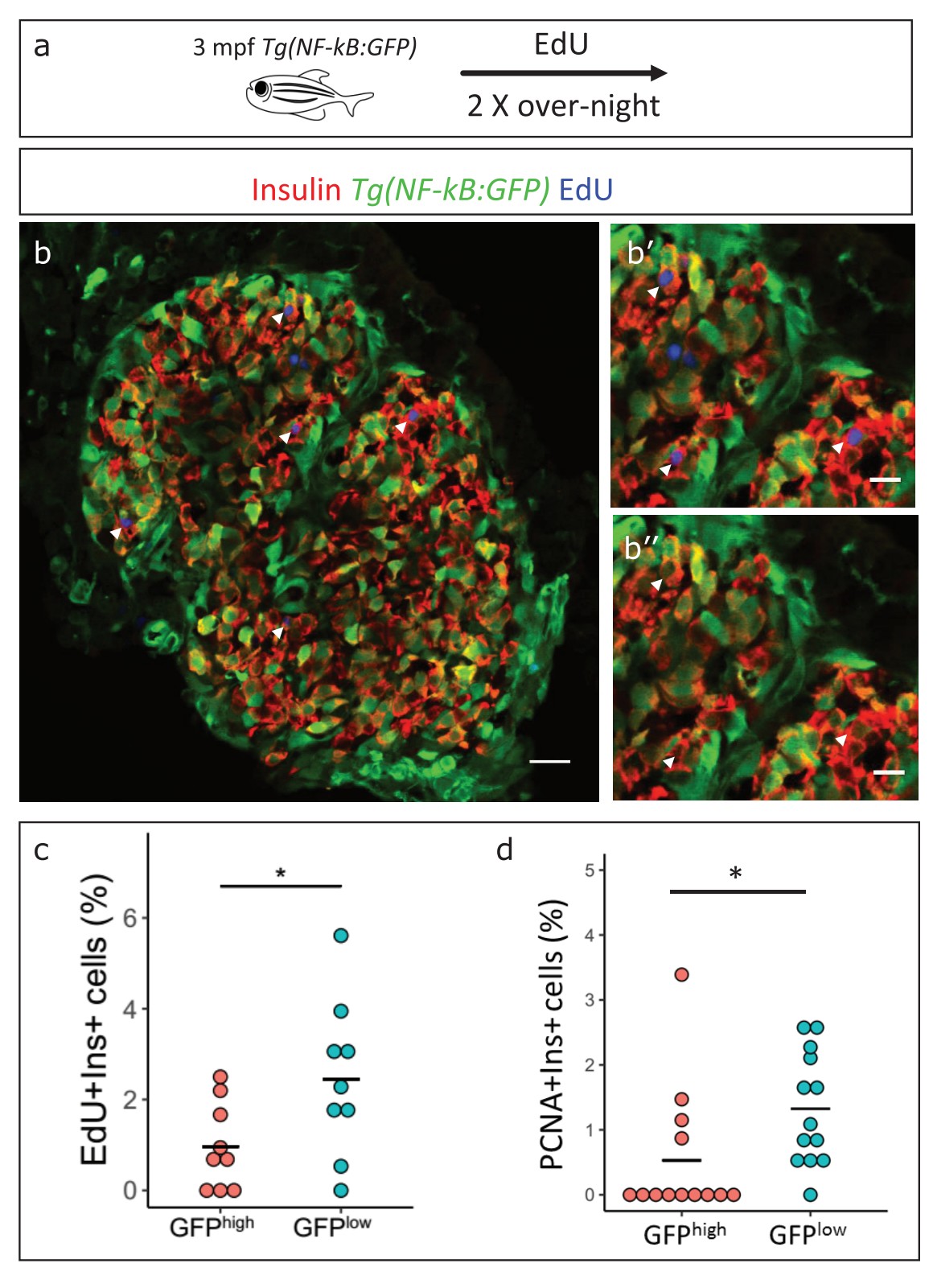

**Figure 5.** *NF-kB*:GFP[high] beta-cells proliferate less than their neighbors. (**a**) Schematic showing the EdU (5-ethynyl-2′-deoxyuridine) incorporation assay. *Tg(NF-kB:GFP)* animals were incubated in EdU at 3 mpf for two consecutive nights and fed during each day. (**b**) EdU incorporation assay was performed to mark the proliferating beta-cells in *Tg(NF-kB:GFP)* animals at 3 mpf. The confocal image (single plane) shows an overview of a section through the islet. Beta-cells were labeled with an insulin antibody (red), a GFP antibody (green) and EdU (blue). Arrowheads point to EdU-positive beta-cells. (**b′–b′′**)
*Figure 5 continued on next page*

*Figure 5 continued*

The insets show higher magnification images with and without the EdU channel. EdU incorporation can be observed in some of the GFP^low cells (white arrow-heads). (c) An insulin-positive cell was first located by going through individual sections in the confocal z-stack. The optical section containing the largest area of the nucleus was chosen as the center of the cell. A region-of-interest (ROI) was drawn around the nucleus and the fluorescence intensities of the GFP and DAPI channels were recorded. The normalized GFP intensity was calculated as a ratio of mean GFP intensity and mean DAPI intensity for each ROI. The average total normalized GFP-intensity of each islet was set as a threshold for dividing the cells into GFP^high and GFP^low populations. The graph shows the percentage of EdU and insulin double-positive cells among the GFP^high and GFP^low populations. Each dot represents one islet (n = 9). Horizontal bars represent mean values (two-tailed t-test, *p<0.05). (d) The graph shows the percentage of PCNA and insulin double-positive cells among the GFP^high and GFP^low populations. Each dot represents one islet (n = 13). Horizontal bars represent mean values (two-tailed t-test, *p<0.05). See also *Figure 5—figure supplement 2* for representative PCNA antibody staining.
DOI: https://doi.org/10.7554/eLife.32965.012

The following figure supplements are available for figure 5:

**Figure supplement 1.** The GFP fluorescence of the transgenic reporter*Tg(NF-kB:GFP)*remains stable for 72 hours in beta-cells.
DOI: https://doi.org/10.7554/eLife.32965.013
**Figure supplement 2.** Proliferating cell nuclear antigen (PCNA) antibody staining shows that *NF-kB*:GF Phighbeta-cells proliferate less than NF-kB:GFP low beta-cells.
DOI: https://doi.org/10.7554/eLife.32965.014

## Discussion

Type two diabetes is an age-related disease, and hence, it is important to identify how advancing age alters the islet. Our work shows that in zebrafish, NF-kB signaling becomes preferentially active in beta-cells from older animals. An additional sign of islet inflammation is the recruitment of intra-islet macrophages, a subset of which express the cytokine *tnfα*. In addition, we show that beta-cells upregulate in a heterogeneous manner the TNFα receptor, *tnfrsf1b*, and that *tnfα*-expression is sufficient to trigger NF-kB signaling activation. Altogether, our results document the development of chronic islet inflammation in older animals. Based on our data, we also propose that with age, beta-cell replication declines in a heterogeneous manner, with high levels of NF-kB signaling marking the cells that lose proliferative potential (*Figure 7*).

The relevance of our results extends beyond the zebrafish model, as they corroborate empirical evidence gathered in human beta-cells. There is emerging evidence that chronic inflammation is a characteristic of aging in human (*Puchta et al., 2016*) and is associated with beta-cell dysfunction in type two diabetes (*Nordmann et al., 2017*). Moreover, the accumulation of innate immune cells in islets in fish is reminiscent of changes observed in type two diabetes in man (*Nordmann et al., 2017*). Furthermore, islets form older human donors exhibit an increase in the number of intra-islet macrophages (*Almaça et al., 2014*), analogous to zebrafish. Thus, our work puts forward the zebrafish as a new model to investigate the mechanisms of beta-cell aging and the crosstalk between beta-cells and the innate immune system, which is of relevance to understanding human disease.

Our work also adds to the burgeoning field of mammalian beta-cell heterogeneity. A recent report from Bonner-Weir and colleagues (*Aguayo-Mazzucato et al., 2017*) revealed progressive increases in the proportion of beta cells expressing age-related markers, including IGF-IR in older mice and human islets, suggesting that aging in mammalian beta-cells might be a heterogeneous process. In our study, we identified a different marker of age-related heterogeneity – NF-kB-activity, and linked this to the proliferative decline of beta-cells, which is an important age-related trait. Intriguingly, the human receptor *TNFRSF11A* (Receptor Activator of NF-kB) shows markedly heterogeneous expression in adult human beta-cells according to the single-cell sequencing database provided by the Sandberg lab (http://sandberg.cmb.ki.se/pancreas/) (*Segerstolpe et al., 2016*). The significance of this observation for human beta-cell heterogeneity needs further investigation. Notably, TNFRSF11A antagonism can increase human beta-cell proliferation, implicating NF-kB signaling in beta-cell proliferation (*Kondegowda et al., 2015*). Indeed, we found that in zebrafish, beta-cells with higher levels of NF-kB signaling elevate *socs2* expression, which in turn can reduce proliferation. Of note, adenovirus transduction of the functionally related gene *socs3* in rat islets inhibits beta-cell proliferation (*Lindberg et al., 2005*). It will be necessary to address whether NF-kB activates *socs2* directly or indirectly to control proliferation.

Recent studies have proposed that beta-cell proliferation and functional maturity exhibit an inverse correlation. For example, gene-expression analysis revealed that proliferating beta-cells

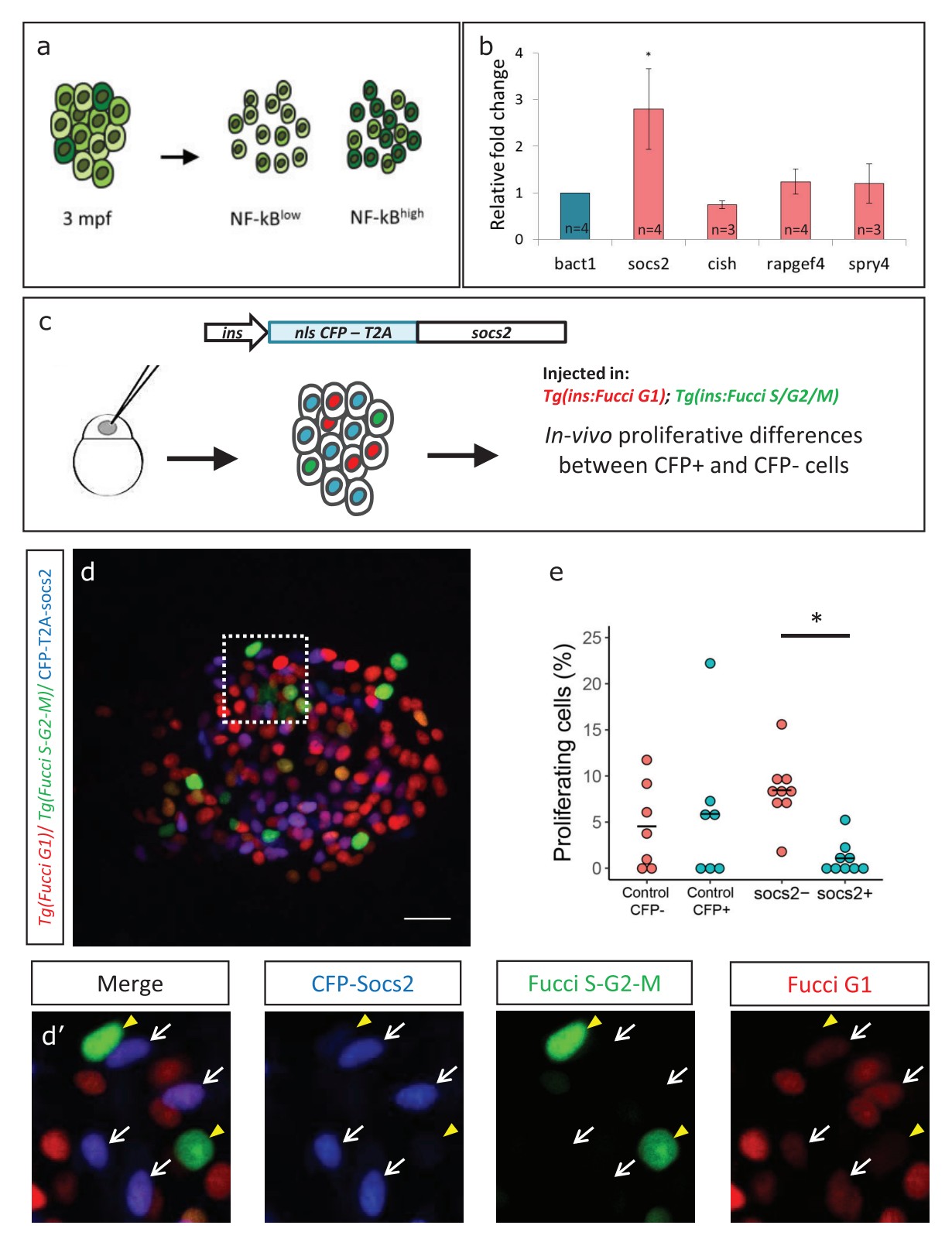

**Figure 6.** *Socs2* is enriched in *NF-kB*:GFP^high cells and inhibits beta-cell proliferation in a cell-autonomous manner. (**a**) Schematic showing the sorting of beta-cells from the double transgenic line *Tg(ins:mCherry);Tg(NF-kB:GFP)* at 3 mpf into GFP^high and GFP^low cells using FACS. (**b**) Bulk RT-qPCR was performed on the GFP^high and GFP^low beta-cells (n = 3 to 4 biological replicates, n = 3 animals per replicate, 1000 cells per condition). Candidate genes significantly enriched in beta-cells at 1 ypf were chosen to be compared between the GFP^high and GFP^low populations at 3 mpf. The graph shows

*Figure 6 continued on next page*

*Figure 6 continued*

relative fold-change between GFP^high and GFP^low cells. The expression of all genes was normalized to β-actin expression before calculating fold-change. *socs2* shows higher expression in the GFP^high cells. Error bars, SD (two-tailed paired t-test, *p<0.05). (c) Schematic showing the method for mosaic overexpression of candidate genes in beta-cells. The *socs2* coding sequence is linked to nuclear-CFP using a T2A sequence. The entire construct was expressed under the insulin promoter. This construct was injected in one-cell-stage-embryos from *Tg(ins:Fucci-G1);Tg(ins:Fucci-S/G2/M)* animals leading to mosaic and stochastic expression of *socs2* in beta-cells during islet development. Control animals were injected with plasmid containing only nuclear-CFP sequence (See **Figure 6—figure supplement 2**). (d) Confocal projections showing mosaic expression of *socs2-T2A-CFP (blue)* at 23 dpf (blue). Proliferating beta-cells are marked by *Tg(ins:Fucci-S/G2/M)* expression (green) and absence of *Tg(ins:Fucci-G1)* expression (red). Anterior to the left. Scale bar 10 μm. (d') Insets show higher magnification single planes from the confocal stacks (white dotted-line in d) with separate channels. The proliferating beta-cells are CFP-negative (yellow arrowheads), whereas some of the non-proliferating cells are CFP-positive (white arrowheads) (e) Quantification of the percentage of *Tg(ins:FUCCI-S/G2/M)*-positive and *Tg(ins:FUCCI-G1)*-negative (green only) beta-cells. The *socs2* expressing β-cells exhibit reduced cell-cycle progression compared to wild-type neighbors (n = 9). Horizontal bars represent mean values (two-tailed t-test, *p<0.05).

DOI: https://doi.org/10.7554/eLife.32965.015

The following source data and figure supplements are available for figure 6:

**Source data 1.** This spreadsheet contains the Relative Fold Change between NF-kB:GFP^high and NF-kB:GFP^low beta-cells used to generate the bar plots and average data shown in **Figure 6b**.

DOI: https://doi.org/10.7554/eLife.32965.018

**Figure supplement 1.** Fluorescent activated cell sorting of *NF-kB*:GFP^high and *NF-kB*:GFP^low beta-cells.

DOI: https://doi.org/10.7554/eLife.32965.016

**Figure supplement 2.** Mosaic expression of candidate genes in beta-cells to study their effect on proliferation.

DOI: https://doi.org/10.7554/eLife.32965.017

reduce the levels of transcripts required for beta-cell function (**Klochendler et al., 2016**). In addition, lineage tracing of immature and mature beta-cells within the same islet revealed higher proliferation of immature beta-cells (**Singh et al., 2017**). In this regard, it will be important to explore whether beta-cells with higher NF-kB signaling are functionally more mature compared to the ones with lower activity, and whether beta-cell function increases in older zebrafish. To start addressing these questions, we performed analysis of beta-cell functional connectivity of our calcium recordings using algorithms developed in the Hodson and Rutter groups (**Hodson et al., 2012**;**Johnston et al., 2016**). However, this analysis did not reveal conclusive changes in beta-cell connectivity with aging (data not shown). In the future, it will be informative to develop new calcium fluorescent reporters allowing to monitor and compare glucose-responsiveness of *NF-kB*:GFP^low and *NF-kB*:GFP^high cells within the same islet.

An intriguing observation of our study is the presence of *tnfα*-positive macrophages in the islets under basal conditions, which might indicate that these macrophages are activated and pro-inflammatory (**Nguyen-Chi et al., 2015**). Under steady state, activated macrophages are typically observed only in barrier organs, such as the lung and the intestine (**Ferris et al., 2017**). Our results now show that activated macrophages are also present in the islets of adult zebrafish under physiological conditions. In agreement with our findings, a recent report documented the presence of islet-resident macrophages expressing TNFα, IL1b and MHC-II (**Ferris et al., 2017**) in non-obese diabetic (NOD) mice. However, despite the presence of cytokine expression in the macrophages, Ferris et al. could not detect nuclear RelA (a member of the NF-kB heterodimer) signaling in the islets, suggesting that beta-cells did not activate NF-kB signaling. This might be a result of the lower sensitivity of their detection method. Indeed, using a sensitive readout of NF-kB signaling based on a transgenic reporter, we show that *tnfα*-expression alone is capable of inducing NF-kB activation in beta-cells. However, we note that further studies and new tools will be necessary to address the crosstalk between the innate immune cells and beta-cell inflammation. Furthermore, additional inflammatory signals can originate from cells other than the macrophages. For example, acinar cells were recently shown to express TNFα, which in turn induces apoptosis in aged mouse beta-cells (**Xiong et al., 2017**).

Besides paracrine factors, NF-kB activity might be regulated by beta-cell intrinsic factors. In particular, ER stress is known to activate NF-kB signaling in multiple cell types (**Tam et al., 2012**). It is possible that with aging, beta-cells experience higher levels of ER stress and thereby activate NF-kB signaling. Therefore, a good question for the future will be to define the contribution of extrinsic and intrinsic factors, including ER-stress, to the heterogeneous activation of inflammation in beta-

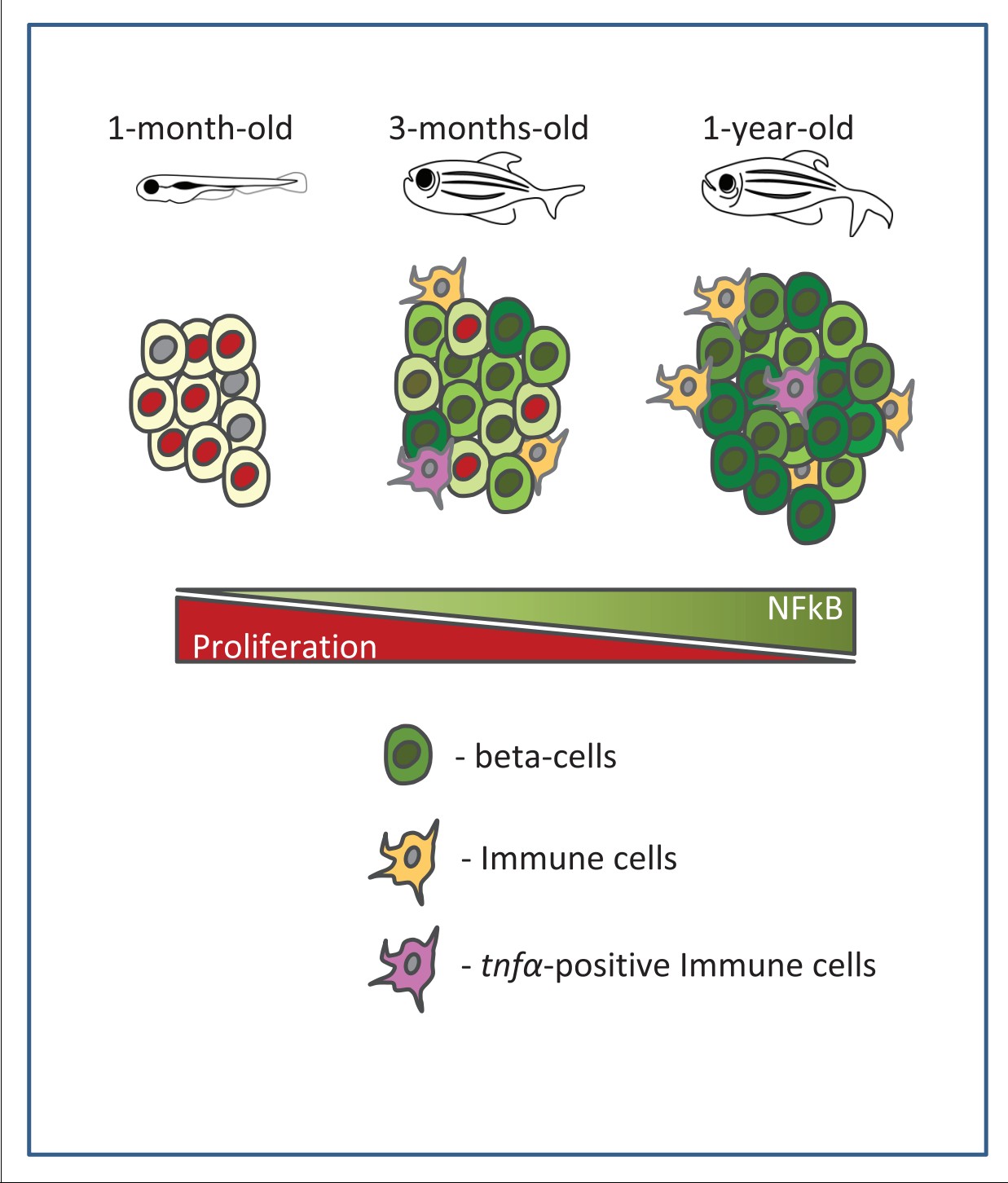

**Figure 7.** A schematic summarizing our model. Beta-cell proliferation declines with age together with a concurrent increase in NF-kB signaling. The activation of NF-kB signaling is heterogeneous among beta-cells and correlates with their proliferative heterogeneity. In particular, beta-cells with higher NF-kB activity proliferate less compared to neighbors with lower activity, and express higher levels of *socs2*, which can inhibit beta-cell proliferation. Furthermore, the crosstalk with *tnfα*-positive immune cells in the islet provides a potential source of inflammation and NF-kB activation in beta-cells.

DOI: https://doi.org/10.7554/eLife.32965.019

cells. Our study opens the possibility to use *Danio rerio* as a new model for gaining insights into the links between aging and beta-cell biology and the relationship between the innate immune system and diabetes.

# Materials and methods

## Key resources table

| Reagent type (species) or resource | Designation | Source or reference | Identifiers | Additional information |
|---|---|---|---|---|
| Gene (*Danio rerio*) | *flag-tnfrsf1b* | synthesized from GenScript | | |
| Gene (*Danio rerio*) | *tnfα* | Dharmacon | MDR1734-202796946 | ZGC tnfa cDNA (CloneId:8148192) |
| Gene (*Danio rerio*) | *cpf-T2A* | synthesized from GenScript | | |
| Genetic reagent (*Danio rerio*) | *Tg(ins:FUCCI-G1)s948* | PMID: 23791726 | | |
| Genetic reagent (*Danio rerio*) | *Tg(ins:FUCCI-S/G2/M)s946* | PMID: 23791726 | | |
| Genetic reagent (*Danio rerio*) | *Tg(NF-kB:GFP)* | PMID: 21439961 | | |
| Genetic reagent (*Danio rerio*) | *TgBAC(tnfα:GFP)* | PMID: 25730872 | | |
| Genetic reagent (*Danio rerio*) | *Tg(ins:nlsRenilla-mKO2)* | PMID: 28939870 | | |
| Genetic reagent (*Danio rerio*) | *Tg(ins:gCaMP6s; cryaa: mCherry)* | PMID: 28939870 | | |
| Genetic reagent (*Danio rerio*) | *Tg(ins: loxP:mCherrySTOP: loxP:H2B-GFP)* | PMID: 21497092 | | |
| Genetic reagent (*Danio rerio*) | *Tg(mpeg1:mCherry)* | PMID: 21084707 | | |
| Genetic reagent (*Danio rerio*) | *Tg(ins:CFP-NTR)* | PMID: 17326133 | | |
| Antibody | anti-insulin | Dako | A0564 | guinea pig (1:200) |
| Antibody | anti-EGFP | Abcam | ab13970 | chicken (1:500) |
| Antibody | anti-PCNA | Dako | M0879 | mouse (1:500) |
| Antibody | anti-L-plastin | Biozol | LS-C210139-250 | rabbit (1:1000) |
| Antibody | Alexa Fluor488, 568 and 647 secondaries | Molecular Probes | | (1:300) |
| Recombinant DNA reagent | *ins:Flag-tnfrsf1b;cryaa:RFP* (plasmid) | This paper | | cloned into *ins:MCS2; cryaa:RFP* |
| Recombinant DNA reagent | *ins:tnfα;cryaa:CFP* (plasmid) | This paper | | cloned by replacing *mCherry-zCdt1* with *tnfα* in *ins:mCherry-zCdt1;cryaa:CFP* |
| Recombinant DNA reagent | *ins:CFP-T2A-socs2;cryaa:RFP* (plasmid) | This paper | | cloned into *ins:MCS2;cryaa:RFP* |
| Recombinant DNA reagent | *ins:CFP-T2A-rapgef4;cryaa:RFP* (plasmid) | This paper | | cloned into *ins:MCS2;cryaa:RFP* |
| Recombinant DNA reagent | *ins:CFP-T2A-spry4;cryaa:RFP* (plasmid) | This paper | | cloned into *ins:MCS2;cryaa:RFP* |
| Recombinant DNA reagent | *ins:mAG-zGeminin;cryaa:RFP* (plasmid) | PMID: 23791726 | | |
| Recombinant DNA reagent | *ins:MCS2;cryaa:RFP* (plasmid) | PMID: 28939870 | | |

*Continued on next page*

Continued

| Reagent type (species) or resource | Designation | Source or reference | Identifiers | Additional information |
|---|---|---|---|---|
| Recombinant DNA reagent | *ins:mCherry-zCdt1;cryaa:CFP* (plasmid) | PMID: 23791726 | | |
| Software, algorithm | edgeR package | PMID:19910308 | | |
| Other | TSQ (N-(6-Methoxy-8-Quinolyl)-p-Toluenesulfonamide) | ThermoFisher | M-688 | 30 μM |

## Zebrafish strains and husbandry

Wild-type or transgenic zebrafish of the outbred TL, AB, WIK strains were used in all experiments. Zebrafish were raised under standard conditions at 28°C. Animals were chosen at random for all experiments. Published transgenic strains used in this study were *Tg(ins:FUCCI-G1)*[s948] (*Ninov et al., 2013*), *Tg(ins:FUCCI-S/G2/M)*[s946] (*Ninov et al., 2013*), *Tg(NF-kB:GFP)* (*Kanther et al., 2011*), *TgBAC(tnfa:GFP)* (*Marjoram et al., 2015*), *Tg(ins:nlsRenilla-mKO2)* (*Singh et al., 2017*), *Tg(ins: loxP: mCherrySTOP:loxP:H2B-GFP)* abbreviated as *Tg(ins:mCherry)* (*Hesselson et al., 2011*), *Tg(mpeg1: mCherry)* (*Ellett et al., 2011*), *Tg(ins:CFP-NTR)* abbreviated as *Tg(ins:CFP)* (*Curado et al., 2007*). Experiments were conducted in accordance with the Animal Welfare Act and with permission of the Landesdirektion Sachsen, Germany (AZ 24–9168, TV38/2015, A12/2016, A5/2017).

## Cloning and constructs

To generate *ins:Flag-tnfrsf1b;cryaa:RFP*, a vector was created by inserting multiple cloning sites (MCS2) downstream of the insulin promoter to yield *ins:MCS2;cryaa:RFP*. To do so, the plasmid *ins: mAG-zGeminin;cryaa:RFP* was digested with EcoRI/PacI and ligated with dsDNA generated by annealing two primers harboring the sites SpeI, BamHI, EcoRV and flanked by EcoRI/PacI overhangs. The plasmid pUC consisting of the *tnfrsf1b* flanked by EcoRI/PacI sites was synthesized from GenScript. Primers were designed such that EcoRI site was destroyed in the process of inserting *tnfrsf1b* under the insulin promoter. *ins:MCS2;cryaa:RFP* and the plasmid *pUC-Flag-tnfrsf1b* were subsequently digested with EcoRI/PacI to yield compatible fragments, which were ligated together to yield the final construct. The entire construct was flanked with I-SceI sites to facilitate transgenesis.

To generate *ins:CFP-T2A-socs2;cryaa:RFP*, a vector was created by inserting multiple cloning sites (MCS2) downstream of the insulin promoter to yield *ins:MCS2; cryaa:RFP*. To do so, the plasmid *ins: mAG-zGeminin;cryaa:RFP* was digested with EcoRI/PacI and ligated with dsDNA generated by annealing two primers harboring the sites SpeI, BamHI, EcoRV and flanked by EcoRI/PacI overhangs. The plasmid pUC consisting of the candidate gene *socs2* fused to CFP via T2A sequence flanked by EcoRI/PacI sites was synthesized from GenScript. Primers were designed such that the EcoRI site was destroyed in the process. *ins:MCS2;cryaa:RFP* and the plasmid *pUC-CFP-T2A-socs2* were subsequently digested with EcoRI/PacI to yield compatible fragments, which were ligated together to yield the final construct. The entire construct was flanked with I-SceI sites to facilitate transgenesis. Same process as described above was used for generating *ins:CFP-T2A-spry4;cryaa:RFP* construct.

To generate *ins:CFP-T2A-rapgef4;cryaa:RFP*, a plasmid pUC consisting of rapgef4 flanked by SpeI/PacI sites was synthesized from GenScript. *ins:CFP-T2A-socs2;cryaa:RFP* and the plasmid *pUC-rapgef4* were subsequently digested with SpeI/PacI to yield compatible fragments, which were ligated together to yield the final construct.

## Analysis of proliferation using mosaic integration in the genome

For counting beta-cells in *Tg(ins:FUCCI-G1);Tg(ins:FUCCI-S/G2/M)* with mosaic expression of candidate genes, the 'spots' function of Imaris (Bitplane) was used after thresholding. The total number of CFP-positive red cells and CFP-negative red cells in the entire islet spanning all stacks were calculated. All the *Tg(ins:FUCCI-S/G2/M)*-positive cells were counted manually for CFP-positive and CFP-negative beta-cells.

$$\text{Percentage of CFP} - \text{positive proliferating cells} = \frac{(\text{CFP}-\text{positive}) + (\text{ins:Fucci}-\text{S/G2/M}-\text{positive and ins:FUCCI}-\text{G1}-\text{negative cells})}{(\text{Total CFP}-\text{positive cells})} \times 100$$

$$\text{Percentage of CFP−negative proliferating cells} =$$
$$\frac{(\text{CFP−negative}) + (\text{ins:Fucci−S/G2/M−positive and ins:FUCCI−G1−negative cells})}{(\text{Total CFP−negative cells})} \times 100$$

## Tissue collection and sectioning

To facilitate confocal imaging of the islet, the pancreas was dissected from the gut (juvenile and adults) after fixation. Fish were killed in Tricaine prior to dissection of gut, and the samples immersed in 4% paraformaldehyde for 48 hr at 4°C. The pancreas was then manually dissected and washed multiple times in PBS.

For cryo-sectioning, the tissue was then immersed in 20% sucrose solution overnight at 4°C. The tissue was then embedded in 20% sucrose +7.5% gelatin solution in cryo-molds on dry ice and sectioned at 14 μm in thickness with Leica cryostat.

## Cell counting

Total number of beta-cells in the islets were counted using Imaris (Bitplane). For counting beta-cells in *Tg(ins:FUCCI-G1);Tg(ins:FUCCI-S/G2/M)*, the 'spots' function of Imaris, with appropriate thresholding, was used to count all the red cells in stacks spanning the entire islet. All the proliferating cells (green only) were counted manually. This approach enabled us to quantify the percentage of proliferating beta-cells in the whole islet.

$$\text{Percentage of proliferating cells} =$$
$$\frac{(\text{ins:Fucci−S/G2/M−positive and ins:FUCCI−G1−negative cells})}{(\text{Total beta−cells})} \times 100$$

## EdU labeling

To label proliferating cells, 3 mpf fish were placed in 2 mM EdU on 2 consecutive nights, and then placed back in system water with normal feeding during each day. The fish were then killed, the gut was fixed and the pancreas was sectioned as described above. The tissue sections were washed 3 × 10 min with PBS, and EdU detection was performed according to the kit protocol Click- iT EdU Alexa Fluor 647 Imaging Kit (C10340 Fisher Scientific). GFP and insulin staining was performed at the concentrations described below.

## Immunofluorescence and image acquisition

Immunofluorescence was performed on pancreas sections prepared as described above. Antigen retrieval was carried out prior to anti-PCNA staining by treating the sections with 10 mM citrate buffer (pH = 6) for 10 mins at 90°C. The sections were permeabilized in 1% PBT (TritonX-100) and blocked in 4% PBTB (BSA). Primary and secondary antibody staining was performed overnight at 4°C. Primary antibodies used in this study were anti-insulin (guinea pig, Dako A0564) at 1:200, anti-EGFP (chicken, abcam ab13970) at 1:500, anti-PCNA (mouse, Dako, M0879) at 1:500, and anti-L-plastin (rabbit, Biozol LS-C210139-250) at 1:1000. Secondary antibodies used in this study were Alexa Fluor 568 and Alexa Fluor 488 anti-guinea pig (1:300), Alexa Fluor 647 anti-rabbit and anti-mouse (1:300) and Alexa Fluor 488 anti-chicken (1:300). Samples were mounted in Vectashield and imaged using a Zeiss LSM 780.

## GCAMP6s image acquisition and analysis

To monitor the changes in glucose-stimulated calcium influx during development, GCAMP6s measurements were performed on isolated islets from *Tg(ins:gCaMP6s; cryaa:mCherry);Tg(ins:Renilla-mKO2; cryaa:CFP)* double-transgenic animals at 3 mpf. Freshly dissected islets were washed with HBSS containing $Ca^{2+}/Mg^{2+}$ (Life technologies, 14175095) twice and embedded in fibrin gels (3:1 ratio of 10 mg/ml Bovine fibrinogen, 50 U/ml Bovine thrombin; Sigma Aldrich, Germany). Upon polymerization, islets were immersed in HBSS containing 2.5 mM glucose and imaged using live confocal microscopy (LSM-780 FLIM inverse) to establish the baseline.

## Fluorescent intensity analysis

Normalized GFP fluorescent intensity of insulin-positive cells on pancreatic islet sections was measured using Fiji (*Schindelin et al., 2012*). An insulin-positive cell was first located by going through individual sections in the confocal z-stack. The optical section containing the largest area of the

nucleus was chosen as the center of the cell. A region-of-interest (ROI) was drawn around the nucleus and the fluorescence intensity of the GFP and DAPI channels were recorded. The normalized GFP intensity was calculated as a ratio of mean GFP intensity and mean DAPI intensity for each ROI. For EdU or PCNA intensity measurements, mean grey intensity value for the EdU or PCNA channel was calculated along with the GFP and DAPI channels in each ROI created at the center of a cell, as described above. To discriminate between GFP<sup>high</sup> and GFP<sup>low</sup> cells, a threshold was set for each islet individually. The threshold (GFP<sup>total</sup>) was calculated as the average normalized GFP intensity of all the images belonging to one islet. Threshold for determining EdU or PCNA-positive cells was set by eye.

The GFP fluorescence intensity of the secondary islets in *Tg(NF-kB:GFP)* animals was calculated with the Imaris software by using the surface function. Surfaces were rendered for each secondary islet using the same threshold. The mean GFP fluorescence intensity and volume within these surfaces was recorded. The GFP fluorescence was normalized to the volume of the secondary islets.

## Cell culture of sorted beta-cells

Beta-cells were dissociated from 3 mpf *Tg(NF-kB:GFP);Tg(ins:mCherry)* islets and FAC-sorted. The single beta-cells were sorted into a 384-well plate, containing the final cell-culture media (50% L-15 (Gibco, 11415–049), 50% DMEM (Gibco, 31966–021), 10% FBS (Gibco, 10500–064) and 1x antibiotics (Sigma, A5955)). The plates were incubated in a cell-culture incubator at 27°C with 5% $CO_2$. Individual beta-cells were imaged using Zeiss LSM-780 inverse confocal microscope. The GFP fluorescence intensity was measured using the ROI function of Fiji as described above.

## FACS and gene profile analysis

For RNA-Seq, RT-qPCR and NF-kB population analysis, beta-cell isolated from islets were sorted and analyzed using FACS-Aria II (BD Bioscience). For dissociation, islets were collected in PBS chilled on ice. After one washing with ice cold PBS, islets were dissociated into single cells by incubation in TrypLE (ThermoFisher, 12563029) with 0.1% Pluronic F-68 (ThermoFisher, 24040032) at 37°C in a benchtop shaker set at 350 rpm for 50 min. Following dissociation, TrypLE was inactivated with 10% FBS, and the cells pelleted by centrifugation at 500 *g* for 10 min at 4°C. The supernatant was carefully discarded and the pellet re-suspended in 500 μl of HBSS (without $Ca^{2+}$, $Mg^{2+}$)+0.1% Pluronic F-68. To remove debris, the solution was passed over a 30 μm cell filter (Miltenyi Biotec, 130-041-407).

For RNA-Sequencing, total RNA was extracted from FACS sorted beta-cells using Quick-RNA MicroPrep kit (R1050 Zymo Research). Sequencing was performed on llumina HiSeq2500 in 2 × 75 bp paired-end mode. Reads were splice-aligned to the zebrafish genome, GRCz10, using GSNAP and known splice sites from Ensembl gene annotation, version 81. FeatureCounts was used to assign reads to exons thus eventually getting counts per gene. EdgeR package of R (*Robinson et al., 2010*) was used to perform differential analysis between samples. Across-samples normalization was performed using the TMM normalization method.

For single-cell RT-qPCR, cDNA was synthesized with Quanta qScript TM cDNA Supermix directly on cells. Total cDNA was pre-amplified for 16 cycles (1 × 95°C 8′, 18x (95°C 45′′, 49°C* 1.30′, 72°C 1.5′) 1 × 72°C 7′) (* with 0.3°C increment/cycle) with the QIAGEN Multiplex PCR Plus Kit (Qiagen) in a final volume of 35 μl in the presence of primer pairs (*Supplementary file 2*, 25 nM final for each primer). Pre-amplified DNA (10 μl) was treated with 1.2 U Exonuclease I and expression quantified by real time PCR on the BioMark HD System (Fluidigm Corporation, CA) using the 96.96 Dynamic Array IFC and the GE 96 × 96 Fast PCR +Melt protocol and SsoFast EvaGreen Supermix with Low ROX (BIO RAD, CA) with 5 μM primers (described above) for each assay. Raw data was analyzed using the Fluidigm Real-Time PCR analysis software.

For bulk RT-qPCR gene expression profiling, 1000 GFP<sup>high</sup> and GFP<sup>low</sup> cells were sorted into 5 μl EB Buffer (Qiagen) containing 0.3% IGEPAL and 0.1% BSA and immediately snap frozen. The cells were then thawed and incubated on ice for 10′. cDNA was synthesized with Quanta qScript TM cDNA Supermix directly on cells in a final volume of 30 μl. 15 μl of cDNA was pre-amplified for 12 cycles (1 × 95°C 1′, 95°C 15′′, 60°C 1′, 72°C 1.5′) and 1 × 72°C 10′ with the TATAA GrandMaster Mix (TATAA Biocenter, Göteborg, Sweden) in a final volume of 35 μl in the presence of primer pairs for the following genes: ins, cish, spry4, trib3, rapgef4, ef1a, bact2, rpl13, tnfa, tnfrsf1b, socs23 (25

nM final for each primer). 1.2 µl pre-amplified cDNA was used for quantification by real time PCR on the LightCycler480 (Roche, Switzerland) using SYBR Premix Ex Taq TM (Tli RNaseH Plus) (Takara BIO USA, INC.) and 0.2 nM of each primer in a volume of 10 µl using the following cycling program: initial denaturation 95°C 30'', amplification 45x (95°C 5'', 60°C 30'') and melting curves 1x (95°C 5'', 60°C 1', ramp to 95°C (ramping rate 0.11)) followed by 30'' cooling at 50°C. Raw data was analyzed using the LightCycler480 analysis software.

For analysing the levels of NF-kB:GFP by FACS, dissociated cells were incubated in 30 µM solution of TSQ (N-(6-Methoxy-8-Quinolyl)-p-Toluenesulfonamide) (ThermoFisher, M-688) for 20 mins to label beta-cells. The cells were pelleted by centrifugation at 500 g for 10 min at 4°C. The supernatant was carefully discarded and the pellet re-suspended in 500 µL of HBSS (without Ca, Mg)+0.1% Pluronic F-68. To remove debris, the solution was passed over a 30 µm cell filter (Miltenyi Biotec, 130-041-407) and proportion of NF-kB:GFP^high and NF-kB:GFP^low cells were analyzed by FACS.

For correlation analysis of GFP fluorescence intensity with GFP mRNA, beta-cells from 3 mpf Tg (ins:mCherry);Tg(NF-kB:GFP) animals were dissociated as described above. Single beta-cells were sorted into 96-well plates using the index sort function of Aria II. This allowed us to record the GFP fluorescence intensity of each sorted beta-cell. Single-cell RT-qPCR was performed on the FAC-sorted cells for GFP and b-actin1 mRNA as described above.

## Analysis of single-cell RT-qPCR data

Single-cell RT-qPCR data was obtained from Fluidigm as Ct values of gene expression per cell. The Fluidigm assay performs 40 cycles of amplification. If the fluorescence signal from RT-qPCR does not cross threshold after 40 cycles, then the gene is considered to be 'not detected', and set as Ct = 40 (*McDavid et al., 2013*). A gene was classified as 'detected' for the value of Ct <40 in a given cell (*McDavid et al., 2013*). Pre-analysis cleanup of the RT-qPCR data was performed by removing cells with undetected values (Ct = 40) for the house keeping genes b-actin1, ef1α or rpl13α. For the beta-cells from 3 mpf and 1 ypf animals, the proportion of cells with detectable candidate gene expression was calculated as:

$$\text{Percentage of cells expressing a candidate gene} = \frac{\text{cells with Ct<40 for the candidate gene}}{\text{Total cells}} \times 100$$

Significance testing for differences in proportion of cells with detectable gene expression was performed using Pearson's Chi-Square test. The Ct values were $-\log_{10}$ transformed for representation purpose, such that $-\log_{10}(40) \sim -1.6$ is considered undetectable gene expression level.

## Statistical analysis

No statistical methods were used to predetermine sample size. The experiments were not blinded. Graphs were plotted using R. Statistical analysis was performed using R and Microsoft Excel. Values were compared using unpaired Students t-test or ANOVA as indicated for each experiment. p-Values of <0.05 were considered statistically significant. Data are expressed as mean ±standard deviation (SD) unless otherwise specified.

## Source data

The raw files and raw count table from deep sequencing can be accessed at GEO with accession number GSE106938.

## Acknowledgements

We thank Max Yun, and members of the Ninov lab for comments on the manuscript, members of CRTD fish, microscopy, Deep Sequencing and FACS facilities for technical assistance. We thank Anne Eugster for assistance with single-cell RT-qPCR and bulk RT-qPCR data. We thank Stephen Renshaw for Tg(NF-kB:GFP) reporter line and Michel Bagnat for TgBAC(tnfα:GFP) reporter line. DJH was supported by a Diabetes UK RD Lawrence (12/0004431) and EFSD/Novo Nordisk Rising Star Fellowships, a Wellcome Trust Institutional Support Award, and an MRC Project Grant (MR/N00275X/1). GAR was supported by Wellcome Trust Senior Investigator (WT098424AIA) and Royal Society Wolfson Research Merit Awards, and by MRC Programmes (MR/J0003042/1 and MR/

R022259/1), MRC Project (MR/N00275X/1), Biological and Biotechnology Research Council (BB/J015873/1) and Diabetes UK Project (11/0004210) grants. This project has received funding from the European Research Council (ERC) under the European Union's Horizon 2020 research and innovation programme (Starting Grant 715884 to DJH). NN is supported by funding from the DFG–Center for Regenerative Therapies Dresden, Cluster of Excellence at TU Dresden and the German Center for Diabetes Research (DZD), as well as research grants from the German Research Foundation (DFG), the European Foundation for the Study of Diabetes (EFSD) and the DZD.

## Additional information

### Funding

| Funder | Grant reference number | Author |
|---|---|---|
| Horizon 2020 Framework Programme | Research and Innovation programme 715884 | David J Hodson |
| DFG-Center for Regenerative Therapies Dresden | | Nikolay Ninov |
| German Center for Diabetes Research | | Nikolay Ninov |
| Deutsche Forschungsgemeinschaft | | Nikolay Ninov |
| European Foundation for the Study of Diabetes | | Nikolay Ninov |

The funders had no role in study design, data collection and interpretation, or the decision to submit the work for publication.

### Author contributions

Sharan Janjuha, Conceptualization, Formal analysis, Methodology, Writing—original draft; Sumeet Pal Singh, Conceptualization, Methodology, Writing—review and editing; Anastasia Tsakmaki, S Neda Mousavy Gharavy, Priyanka Murawala, Judith Konantz, Sarah Birke, David J Hodson, Guy A Rutter, Gavin A Bewick, Methodology; Nikolay Ninov, Conceptualization, Supervision, Funding acquisition, Writing—original draft, Project administration

### Author ORCIDs

Sharan Janjuha  http://orcid.org/0000-0002-5910-2912
Sumeet Pal Singh  http://orcid.org/0000-0002-5154-3318
David J Hodson  http://orcid.org/0000-0002-8641-8568
Guy A Rutter  https://orcid.org/0000-0001-6360-0343
Gavin A Bewick  https://orcid.org/0000-0002-4335-8403
Nikolay Ninov  http://orcid.org/0000-0003-3286-6100

### Ethics

Animal experimentation: Experiments were conducted in accordance with the Animal Welfare Act and with permission of the Landesdirektion Sachsen, Germany (AZ 24-9168, TV38/2015, A12/2016, A5/2017).

### Decision letter and Author response

Decision letter https://doi.org/10.7554/eLife.32965.026
Author response https://doi.org/10.7554/eLife.32965.027

## Additional files

### Supplementary files

• Supplementary file 1. List of genes differentially expressed from RNA-Seq of beta-cells at 3 mpf and 1 ypf ($\log_2$FC ± 1.5).
DOI: https://doi.org/10.7554/eLife.32965.020

• Supplementary file 2. List of primer sequences of genes validated using single-cell RT-qPCR and bulk RT-qPCR.
DOI: https://doi.org/10.7554/eLife.32965.021

• Transparent reporting form
DOI: https://doi.org/10.7554/eLife.32965.022

### Major datasets

The following dataset was generated:

| Author(s) | Year | Dataset title | Dataset URL | Database, license, and accessibility information |
|---|---|---|---|---|
| Sharan Janjuha, Sumeet Pal Singh, Nikolay Ninov | 2018 | Age-related Islet Inflammation Marks the Proliferative Decline of Pancreatic Beta-cells in Zebrafish | https://www.ncbi.nlm.nih.gov/geo/query/acc.cgi?acc=GSE106938 | Publicly available at the NCBI Gene Expression Omnibus (accession no: GSE106938). |

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
