## [Decision Letter]

Thank you for sending your article entitled "Aging is a heterogeneous process across individual beta-cells" for peer review at *eLife*. Your article has been evaluated by three peer reviewers, and the evaluation is overseen by Marianne Bronner as the Senior and Reviewing Editor.

The manuscript by Janjuha et al. provides information about the aging process in zebrafish beta cells. Beautiful in vivo genetic imaging tools allow visualization of glucose-mediated calcium uptake, proliferation, cell cycle stage, and NF-κB activity over a significant period of growth of the zebrafish pancreas. The data suggest that increasing NF-κB activity in beta cells with age, possibly driven by resident macrophages expressing TNF-α, is associated with increased Socs2 expression, which may inhibit proliferation. The methods are mostly sound, the imaging and quantification are of high quality.

The most salient issue was that the reviewers were not convinced that the data revealed substantial new findings that could be definitively seen as moving the field forward beyond what has already been determined by a plethora of studies in various mammalian cell line and model animal systems. In another example, how likely is the result from studying primarily the large principal islet likely to represent zebrafish-specific findings? Therefore, we ask you to address how and why these results inform upon novel aspects of beta-cell biology that would be applicable to mammalian systems or to islet pathology? How does this study help understand the role of aging, or its potential impact on diabetogenesis?

For a general readership as expected for *eLife*, the journal is interested in providing a definable step forward, rather than reiterating what might essentially already be known, albeit defined using novel, attractive cell-biological tools.

Major Comments:

1) The individual cell analysis is based on single-cell qPCR, a technique that may not be robust enough to support this type of conclusion. The data in 2D appear to be binary, the *tnfrsf1b* data in 2D don't agree well with the sensitive imaging/flow data in 3D-G, and reference is made in the Materials and methods to cells that failed to amplify housekeeping genes. Please provide evidence of the robustness of this technique.

2) The EdU label was over a prolonged period, but the NF-κB reporter reflects activity at time of death only. Please provide evidence of stability over time of NF-κB reporter (is this a stable cellular characteristic, or are the cells fluxing in/out of NF-κB activity?) or repeat the experiment with a proliferation marker that labels cells dividing at the time of death. Alternatively, were proliferation markers decreased in the FACS-sorted GFP-high population?

3) The principal islet is the only islet shown here, and either extension to include the secondary islets that emerge later should be included, or rationale as to why this is irrelevant or inappropriate should be considered.

4) The heterogeneity concept is too strongly emphasized in the title and Abstract. The Abstract (and Discussion) suggest the conclusions are based on single cell seq, but this experiment is not included in this manuscript. The work stands without the forced heterogeneity angle. The authors should soften these statements so the main points can be better projected.

5) The Discussion should address (more) explicitly to what degree these results might or might not be (zebra)fish-specific. Also, the Discussion is quite long, and should be reorganized into separate sub-themes.

6) A major issue to be dealt with (directly by some evidence), or at least discussed, is the contribution or not of GFP longevity to some of the reporter transgenes used.

7) Figure 6B lacks significance labels.

---

## [Author Response]

*The manuscript by Janjuha et al. provides information about the aging process in zebrafish beta cells. Beautiful* in vivo genetic imaging tools allow visualization of glucose-mediated calcium uptake, proliferation, cell cycle stage, and NF-κB activity over a significant period of growth of the zebrafish pancreas. The data suggest that increasing NF-κB activity in beta cells with age, possibly driven by resident macrophages expressing TNF-α, is associated with increased Socs2 expression, which may inhibit proliferation. The methods are mostly sound, the imaging and quantification are of high quality.The most salient issue was that the reviewers were not convinced that the data revealed substantial new findings that could be definitively seen as moving the field forward beyond what has already been determined by a plethora of studies in various mammalian cell line and model animal systems.

To the best of our knowledge, the results of the study have not been previously shown in mammalian cell lines and model animal systems.

1) Our study is the first to show that differential NF-κB signaling marks beta-cell populations based on their proliferative status during aging.

2) We show for the first time that NF-κB signaling can distinguish young and old beta-cells, acting as a marker of advancing beta-cell age.

3) We strengthen the new concept of heterogeneous beta-cell aging, which was also proposed for mouse and human beta-cells by Bonner-Weir and colleagues (discussed below). In addition, we identify new markers of this heterogeneity.

4) We put forward the zebrafish as a new model to investigate the mechanisms of beta-cell aging and the crosstalk between beta-cells and the innate immune system, which is of relevance to understanding human disease

In another example, how likely is the result from studying primarily the large principal islet likely to represent zebrafish-specific findings?

Several pieces of evidence argue in favor of our findings applying to human beta-cells.

1) To further address the relevance of our results to human beta-cells, we performed additional in-silico analysis of human data. For this, we re-analyzed the single-cell RNA sequencing data from aging human beta-cells (Enge et al., 2017). We found that a total of 4387 genes were significantly up-regulated with advancing age in their data set, including genes involved in the NF-κB pathway, such as TNFRSF1A, TNFRSF21, TNFRSF19 and NFKB1. We compared all the upregulated genes from Enge et al. to a published database of NF-κB target genes generated by the Gilmore Lab at Boston University (https://www.bu.edu/NF-κB/gene-resources/targetgenes/). We found 52 NF-κB target genes that are upregulated with age in the Enge et al. dataset.

This correlation is significant (Chi-square test with Yates correction, p < 0.001). Overall, this insilico analysis hints towards increased NF-κB signaling with age in human beta-cells. In addition, there is previous evidence for increasing inflammation and infiltration by macrophages in human islets with age (Almaça et al., 2014), as in our zebrafish model. Thus, it is likely that our model is relevant and will inform future work in man.

2) Regarding the contribution of our work to the burgeoning field of beta-cell heterogeneity, a recent report from Bonner-Weir and colleagues (Aguayo-mazzucato et al., 2017) revealed progressive increases in the proportion of beta cells expressing IGF-IR with age in mice and human, suggesting that aging in human beta-cells might also be a heterogeneous process. In our study, we identified a different marker of age-related heterogeneity – NF-κB, and linked this to the proliferative status of the cells, which is an important age-related trait.

3) Supporting the relevance of our results to beta-cells in man, the human receptor TNFRSF11A shows markedly heterogeneous expression in adult human beta-cells according to the single-cell sequencing database from the Sandberg lab

(http://sandberg.cmb.ki.se/pancreas/) (Segerstolpe et al., 2016). The significance of this observation for human beta-cell heterogeneity needs further investigation. Our work warrants such investigation. It is important to note that TNFRSF11A antagonism can increase human beta-cell proliferation, consistent with an involvement of NF-κB signaling in this process (Kondegowda et al., 2015).

Therefore, we ask you to address how and why these results inform upon novel aspects of beta-cell biology that would be applicable to mammalian systems or to islet pathology? How does this study help understand the role of aging, or its potential impact on diabetogenesis?

There is emerging evidence that chronic inflammation is a characteristic of aging in human (Puchta et al., 2016) and is associated with beta-cell dysfunction in type 2 diabetes (Nordmann et al., 2017). Moreover, the accumulation of innate immune cells in islets in fish is reminiscent of changes observed in type 2 diabetes in man (Nordmann et al., 2017). Furthermore, islets form older human donors exhibit an increase in the number of intra-islet macrophages (Almaça et al., 2014), analogous to zebrafish. Since our study is the first of this kind in zebrafish, it opens the possibility to use *Danio rerio* as a new model for gaining insights into the links between aging and beta-cell biology and the relationship between the innate immune system and diabetes.

For a general readership as expected for eLife, the journal is interested in providing a definable step forward, rather than reiterating what might essentially already be known, albeit defined using novel, attractive cell-biological tools.Major Comments:1) The individual cell analysis is based on single-cell qPCR, a technique that may not be robust enough to support this type of conclusion. The data in 2D appear to be binary, the tnfrsf1b data in 2D don't agree well with the sensitive imaging/flow data in 3D-G, and reference is made in the Materials and methods to cells that failed to amplify housekeeping genes. Please provide evidence of the robustness of this technique.

There is emerging evidence that chronic inflammation is a characteristic of aging in human (Puchta et al., 2016) and is associated with beta-cell dysfunction in type 2 diabetes (Nordmann et al., 2017). Moreover, the accumulation of innate immune cells in islets in fish is reminiscent of changes observed in type 2 diabetes in man (Nordmann et al., 2017). Furthermore, islets form older human donors exhibit an increase in the number of intra-islet macrophages (Almaça et al., 2014), analogous to zebrafish. Since our study is the first of this kind in zebrafish, it opens the possibility to use *Danio rerio* as a new model for gaining insights into the links between aging and beta-cell biology and the relationship between the innate immune system and diabetes.

2) The EdU label was over a prolonged period, but the NF-κB reporter reflects activity at time of death only. Please provide evidence of stability over time of NF-κB reporter (is this a stable cellular characteristic, or are the cells fluxing in/out of NF-κB activity?) or repeat the experiment with a proliferation marker that labels cells dividing at the time of death. Alternatively, were proliferation markers decreased in the FACS-sorted GFP-high population?

a) To provide evidence that the NF-κB reporter is stable in the low and high populations, we applied time-lapse imaging of individual beta-cells in culture. The EdU incorporation was performed over the course of 48 hours. To ensure that the NF-κB reporter (GFP) was stable during this time, we FAC-sorted GFP^high^ and GFP^low^ beta-cells from *Tg(NF-κB:GFP);Tg(ins:mCherry)* 3 mpf animals. In order to confirm that the GFP fluorescence of beta-cells remains stable over the two-day period of EdU incorporation, we followed individual FAC-sorted *NF-κB*:GFP^high^ and ^low^ beta-cells over 72 hours ex vivo. Indeed, the GFP fluorescence remained stable over the time-period of the experiment (Figure 5—figure supplements 1A-D). The GFP^high^ cells retained their GFP signal and the GFP^low^ cells did not turn on the GFP signal.

b) As an alternative to EdU labeling, which required incubation over long periods, we performed Proliferating cell nuclear antigen (PCNA) antibody staining in sections from *Tg(NF-κB:GFP);Tg(ins:mCherry)* animals. PCNA assay allowed us to label proliferating cells at the time of death, as suggested by the reviewers. The result presented in Figure 5—figure supplement 2 and Figure 5D, shows a higher proportion of NF-κB:GFP^low^ cells to be positive for the proliferation marker PCNA, as compared to NF-κB:GFP^high^ cells. The PCNA assay corroborates and strengthens our result that NF-κB:GFP^low^ beta-cells are more proliferative than NF-κB:GFP^high^ cells.

3) The principal islet is the only islet shown here, and either extension to include the secondary islets that emerge later should be included, or rationale as to why this is irrelevant or inappropriate should be considered.

We have extended our analysis to include secondary islets. This analysis documented the decline in beta-cell proliferation within the secondary islets (Figure 1—figure supplement 1), and increased activation of NF-κB signaling with age in the secondary islets (Figure 3—figure supplement 1). These results, combined with the analysis of the primary islets, show the applicability of our findings for the entire zebrafish beta-cell population.

4) The heterogeneity concept is too strongly emphasized in the title and Abstract. The Abstract (and Discussion) suggest the conclusions are based on single cell seq, but this experiment is not included in this manuscript. The work stands without the forced heterogeneity angle. The authors should soften these statements so the main points can be better projected.

We softened the statements to give a more balanced projection of the main points, as suggested by the reviewers. In line with these changes, we have changed the title of our study to: “Age-related Islet Inflammation Marks the Proliferative Decline of Pancreatic Beta-cells in Zebrafish”. Furthermore, we re-wrote the Abstract and the Discussion.

5) The Discussion should address (more) explicitly to what degree these results might or might not be (zebra)fish-specific. Also, the Discussion is quite long, and should be reorganized into separate sub-themes.

As suggested by the reviewers, we have drastically shortened the Discussion (from 6 to 3 pages), while making it more focused. Specifically, we discussed the relevance of our study to the mammalian models, and also explicitly indicated how the study adds to the existing knowledge. We hope the reviewers find our revised Discussion to be improved. To avoid losing citations of important previous literature, we moved some of the citations to the Introduction.

6) A major issue to be dealt with (directly by some evidence), or at least discussed, is the contribution or not of GFP longevity to some of the reporter transgenes used.

We agree with the reviewers that the GFP-protein life-span could contribute to the NF-κB reporter activity. We took this concern seriously, and we addressed it using two approaches:

a) As described in point 1 above, we measured the expression levels of GFP mRNA in relation to GFP-fluorescence intensity in individual beta-cells. The result, presented in Figure 3—figure supplement 3B, shows a positive (R^2^ = 0.28) correlation between GFP-mRNA level and GFP-fluorescence intensity in individual beta-cells. The positive correlation points to a higher transcription from the NF-κB reporter in the GFP-high beta-cells. This in turn suggests a stronger activation of the NF-κB signaling in the GFP-high cells.

b) Using qPCR, we quantified the differences in GFP mRNA levels between beta-cells from 3 mpf and 1 ypf islets. To this end, we isolated beta-cell using the *Tg(NF-κB:GFP); Tg(ins:mCherry)* lines, and measured the mRNA levels of GFP and beta-actin1 (housekeeping gene). The qPCR results, presented in Figure 3—figure supplement 3A, show 50% increase in GFP mRNA expression in beta-cells from 1 ypf islets as compared to beta-cells from 3 mpf islets. These data corroborate the increase in NF-κB reporter activity in beta-cells between the two time points (Figure 3).

7) Figure 6B lacks significance labels.

We apologize for the missing significance labels. We corrected this.

References:

Enge M, Arda HE, Mignardi M, Beausang J, Bottino R, Kim SK & Quake SR (2017). Single-Cell Analysis of Human Pancreas Reveals Transcriptional Signatures of Aging and Somatic Mutation Patterns. Cell 171: 321–330.e14 Available at: https://doi.org/10.1016/j.cell.2017.09.004

Papalexi E & Satija R (2017) Single-cell RNA sequencing to explore immune cell heterogeneity. Nat. Rev. Immunol. 18: 35–45 Available at: http://www.nature.com/doifinder/10.1038/nri.2017.76